# On Trace of PGD-Like Adversarial Attacks

## Abstract

Adversarial attacks pose safety and security concerns for deep learning applications. Yet largely imperceptible, a strong PGD-like attack may leave strong trace in the adversarial example. Since attack triggers the local linearity of a network, we speculate network behaves in different extents of linearity for benign examples and adversarial examples. Thus, we construct *Adversarial Response Characteristics* (ARC) features to reflect the model's gradient consistency around the input to indicate the extent of linearity. Under certain conditions, it shows a gradually varying pattern from benign example to adversarial example, as the later leads to *Sequel Attack Effect* (SAE). ARC feature can be used for *informed* attack detection (perturbation magnitude is known) with binary classifier, or *uninformed* attack detection (perturbation magnitude is unknown) with ordinal regression. Due to the uniqueness of SAE to PGD-like attacks, ARC is also capable of inferring other attack details such as loss function, or the ground-truth label as a post-processing defense. Qualitative and quantitative evaluations manifest the effectiveness of ARC feature on CIFAR-10 w/ ResNet-18 and ImageNet w/ ResNet-152 and SwinT-B-IN1K with considerable generalization among PGD-like attacks despite domain shift. Our method is intuitive, light-weighted, non-intrusive, and data-undemanding.

## 1  Introduction

Recent studies have revealed the vulnerabilities of deep neural networks by adversarial attacks [1, 2], where undesired output (*e.g.* misclassification) could be incurred by an imperceptible perturbation added to network input, posing safety and security concerns for respective applications. In the literature, PGD-like attacks, including BIM [1], PGD [2], MIM [3], and APGD [4], are strong and widely used. Yet, such strong attack may also leave strong trace in its result, as does in the feature maps [5]. Consider an *extremely limited setting* – given an *already trained* deep neural network and merely a *tiny* set (*e.g.*, 50) of training data, *without* any change in architecture or weights, *nor* any auxiliary deep networks, can we still identify any trace of adversarial attack?

Recall that FGSM [6], the foundation of PGD-like attacks, attributes network vulnerability to "local linearity" being easily triggered by adversarial perturbations. Thus, we conjecture that a network behaves in a higher extent of linearity to adversarial examples than to benign (*i.e.*, unperturbed) ones. With the first-order Taylor expansion of a network, "local linearity" implies high gradient proximity in the respective local area. Thus, we can select a series of data points with stable pattern near the input as exploitation vectors using BIM [1] attack, and then compute the model's Jacobian matrices with respect to them. Next, the *Adversarial Response Characteristics* (ARC) matrix is constructed from these Jacobian matrices reflecting the gradient direction consistency across all exploitation vectors. Different from benign examples, PGD-like attacks will trigger *Sequel Attack Effect* (SAE), leaving higher values in the ARC matrix and hence reflecting higher gradient consistency among exploitation vectors around the input. Visualization results suggest SAE is a gradually varying pattern with perturbation magnitude increasing, indicating feasibility of attack detection.

Submitted to 36th Conference on Neural Information Processing Systems (NeurIPS 2022). Do not distribute.

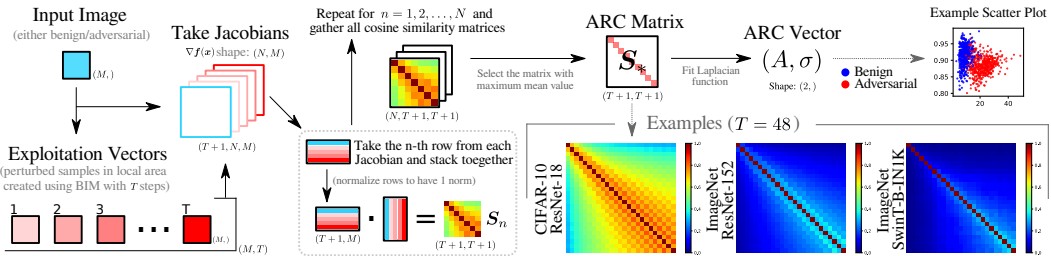

Figure 1: Diagram for computing the ARC matrix and the ARC vector. They reflect the model's gradient consistency within a local linear area around the input to indicate the extent of linearity. Shallow network like ResNet-18 shows higher linearity to benign examples, while deeper networks like ResNet-152 and SwinT-B-IN1K show lower linearity.

The ARC matrix can be simplified into the 2-D ARC vector by fitting a Laplacian function due to their resemblance, in order to make subsequent procedure simple to interpret. The ARC vector can be used for *informed* attack detection (the perturbation magnitude $\varepsilon$ is known) with an SVM-based binary classifier; or for *uninformed* attack detection (the perturbation magnitude $\varepsilon$ is unknown) with an SVM-based ordinal regression model. The SAE is the unique trace of PGD-like attacks. Due to the uniqueness of SAE to PGD-like attacks, once the attack is detected, we can also infer some attack details including the attack loss function, or the ground-truth label used during the attack as a post-processing defense method.

We evaluate our method on CIFAR-10 [7] with ResNet-18 [8], and ImageNet [9] with ResNet-152 [8] and SwinT-B-IN1K [10]. Qualitative and quantitative experimental results manifest the effectiveness of our method in identifying SAE, the unique trace of PGD-like attacks for attack detection, which also possess considerable generalization capability (despite domain shift among PGD-like attacks) even if training data only involves few benign and adversarial examples from BIM attack.

**Contributions.** We present the ARC features to identify the unique trace, *i.e.*, SAE of PGD-like attacks from adversarially perturbed inputs. It can be used for informed/uninformed attack detection and inferring attack details (including correcting prediction). Through the lens of ARC feature (reflecting network's gradient behavior), we also obtain insights on why networks are vulnerable, as well as why adversarial training works well as a defense. Although our method is only sensitive to PGD-like attacks, it is (1) light-weighted (requires no auxiliary deep model); (2) non-intrusive (requires no change to the network architecture or weights); (3) data-undemanding (can generalize with very few samples). Such a problem setting is extremely limited, requiring strong cues to solve.

## 2 Adversarial Response Characteristics & Sequel Attack Effect

A neural network $\boldsymbol{f}(\cdot)$ maps the input $\boldsymbol{x} \in \mathbb{R}^M$ into a pre-softmax output $\boldsymbol{y} \in \mathbb{R}^N$, where the maximum element after softmax corresponds to the class prediction $\hat{c}(\boldsymbol{x})$, which is expected to match with the ground truth $c(\boldsymbol{x})$. Then, a typical adversarial attack [1, 2] aims to find an imperceptible adversarial perturbation $\boldsymbol{r} \in \mathbb{R}^M$ that induces misclassification, *i.e.*, $\arg\max_n f_n(\boldsymbol{x} + \boldsymbol{r}) \neq c(\boldsymbol{x})$ where $\|\boldsymbol{r}\|_p \leq \varepsilon$, $\boldsymbol{x} + \boldsymbol{r} \in [0, 1]^M$, and $f_n(\cdot)$ is the $n$-th element of vector function $\boldsymbol{f}(\cdot)$.

According to FGSM [6], the neural network is vulnerable because the "locally linear" property being triggered by the attack. Thus, we assume that the neural network $\boldsymbol{f}(\cdot)$ behaves relatively non-linear against benign examples, while relatively linear against adversarial examples. Then, $\boldsymbol{f}(\cdot)$ can be approximated by the first-order Taylor expansion around an either benign or adversarial sample $\tilde{\boldsymbol{x}}$:

$$\tilde{\boldsymbol{x}} \triangleq \boldsymbol{x} + \boldsymbol{r}, \quad f_n(\tilde{\boldsymbol{x}} + \boldsymbol{\delta}) \approx f_n(\tilde{\boldsymbol{x}}) + \boldsymbol{\delta}^T \nabla f_n(\tilde{\boldsymbol{x}}), \quad \forall n \in \{1, 2, \ldots, N\}, \tag{1}$$

where $\boldsymbol{\delta}$ is a small vector exploiting the local area around the point $\tilde{\boldsymbol{x}}$, and the gradient vector $\nabla f_n(\cdot)$ is the $n$-th row of the Jacobian $\nabla \boldsymbol{f}(\cdot)$ of size $N \times M$. We name the twice-perturbed $\tilde{\boldsymbol{x}} + \boldsymbol{\delta}$ as "exploitation vector". This equation means in order to reflect linear behaviour, the first-order gradient $\nabla f_n(\cdot)$ is expected to remain in high consistency (or similarity) in the local area regardless of $\boldsymbol{\delta}$. In contrast, when the input $\tilde{\boldsymbol{x}}$ is not adversarial ($\boldsymbol{r} = \boldsymbol{0}$), neither Taylor approximation nor the gradient consistency is expected to hold. Next, the gradient consistency will be quantized to verify our conjecture, and reveal difference between benign and adversarial inputs.

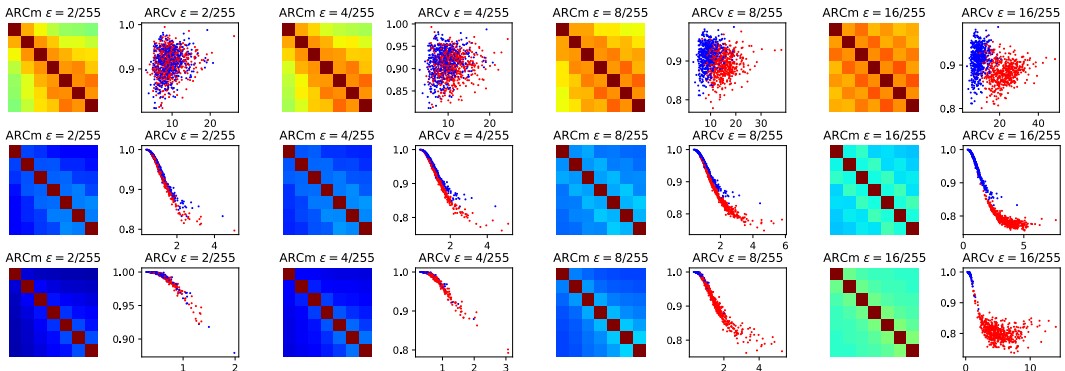

Figure 2: The ARC features (*i.e.* ARC matrix/vector) of adversarial examples created by the BIM attack. 1st row: ResNet-18 on CIFAR-10; 2nd row: ResNet-152 on ImageNet; 3rd row: SwinT-B-IN1K on ImageNet. Blue and red dots in the scatter plots correspond to the benign and adversarial examples, respectively. The cluster centers of the ARC vector correlates with the perturbation magnitude $\varepsilon$.

**Adversarial Response Characteristics (ARC).** Using random noise as $\boldsymbol{\delta}$ does not lead to a stable pattern of change in a series of exploitation vectors $\{\tilde{\boldsymbol{x}} + \boldsymbol{\delta}_t\}_{t=0,1,\ldots,T}$. Instead, we use Basic Iterative Method (BIM) [1] to make $\boldsymbol{f}(\cdot)$ more linear starting from $\tilde{\boldsymbol{x}}$, which means to "continue" the attack if $\tilde{\boldsymbol{x}}$ is already adversarial, or "restart" otherwise. However, the ground-truth label for an arbitrary $\tilde{\boldsymbol{x}}$ is *unknown*. Since PGD-like attacks tend to make the ground-truth least-likely based on our observation, we treat the least-likely prediction $\check{c}(\boldsymbol{x})$ as the label. Then, the BIM iteratively maximizes the cross entropy loss $L_{\text{CE}}(\tilde{\boldsymbol{x}} + \boldsymbol{\delta}, \check{c}(\boldsymbol{x}))$ via projected gradient ascent as

$$\boldsymbol{\delta}_{t+1} \leftarrow \text{Clip}_{\Omega}\Big(\boldsymbol{\delta}_t + \alpha \, \text{sign}[\nabla L_{\text{CE}}(\tilde{\boldsymbol{x}} + \boldsymbol{\delta}_t, \check{c}(\boldsymbol{x}))]\Big), \quad t = 1, 2, \ldots, T, \tag{2}$$

where $\text{Clip}_{\Omega}(\cdot)$ clips the perturbation to the $L_p$ bound centered at $\tilde{\boldsymbol{x}}$, and $\boldsymbol{\delta}_0 = \boldsymbol{0}$. If the input $\tilde{\boldsymbol{x}}$ is benign, then the network behaviour is expected to changed from "very non-linear" to "somewhat-linear" during the process; if the input $\tilde{\boldsymbol{x}}$ is already adversarially perturbed, then the process will "continue" the attack, making the model even more "linear" – we call this *Sequel Attack Effect* (SAE).

To quantize the extent of "linearity", we measure the model's gradient consistency across exploitation vectors with cosine similarity. For each $f_n(\cdot)$, we construct a matrix $\boldsymbol{S}_n$ of shape $(T+1, T+1)$:

$$s_n^{(i,j)} = \cos\big[\nabla f_n(\tilde{\boldsymbol{x}} + \boldsymbol{\delta}_i), \nabla f_n(\tilde{\boldsymbol{x}} + \boldsymbol{\delta}_j)\big], \quad \forall i, j = 0, 1, \ldots, T. \tag{3}$$

As the model $\boldsymbol{f}(\cdot)$ becoming more "linear" to the input (higher gradient consistency), the off-diagonal values in $\boldsymbol{S}_n$ is expected to gradually increase from the top-left to the bottom-right corner. Note that the attack may not necessarily make all $f_n(\cdot)$ behave linear, so we select the most representative cosine matrix with the highest mean as the *ARC matrix*: $\boldsymbol{S}_* \triangleq \boldsymbol{S}_{n^*}$, where $n^* = \arg\max_n \sum_{i,j} s_n^{(i,j)}$.

Due to the resemblance of the ARC matrix to the Laplacian function with matrix diagonal being the center, we simplify it into a two-dimensional *ARC vector* $(A, \sigma)$ by fitting $\mathcal{L}(i, j; A, \sigma) = A \exp(-|i - j|/\sigma)$ with Levenberg-Marquardt algorithm [11], where $i, j$ are matrix row and column indexes, while $A$ and $\sigma$ are function parameters. For brevity, we abbreviate the ARC matrix as "ARCm", and the ARC vector as "ARCv". The process for computing them is summarized in Fig. 1.

**Visualizing Sequel Attack Effect (SAE).** We compute ARCm based on some benign examples using $T=48$, as shown in Fig. 1. The trend of being gradually "linear" (higher cosine similarity) along the diagonal is found across architectures. Thus, SAE is similar to "continue" attack from halfway on the diagonal in such a large ARCm. As illustrated in Fig. 2, already adversarially perturbed input (using BIM) leads to larger cosine similarity at the very first exploitation vectors as perturbation magnitude $\varepsilon$ increases from 0 to 16/255. Meanwhile, the cluster separation for ARCv is more and more clear. Thus, a clear and gradually changing pattern can be seen in ARCm and ARCv. This pattern is even valid and clear for the state-of-the-art ImageNet models. In brief, SAE is reflected by higher gradient consistency in ARCm, or greater $\sigma$ and smaller $A$ in ARCv. Similar visualization from other PGD-like attacks, including PGD [2], MIM [3] and APGD [4] in Fig. 3, indicates the possibility of generalization for all PGD-like attacks with only training samples from the BIM attack despite domain shift. We adopt SVM afterwards to retain explainability and simplicity.

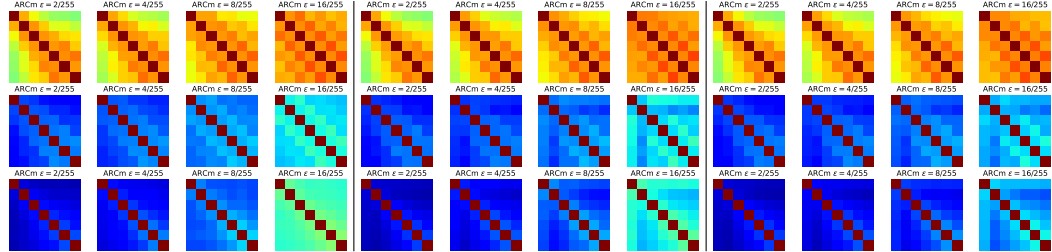

Figure 3: ARCm with adversarial examples created by PGD (left), MIM (middle), and APGD (right) attacks. The three rows correspond to ResNet-18, ResNet-152, and SwinT-B-IN1K, respectively. It is clear that PGD-like attacks qualitatively manifest similar SAE through ARCm.

**Uniqueness of SAE to PGD-Like Attack.** Whether SAE can be consistently triggered depends on whether the following conditions are simultaneously *true*: (I) whether the input is adversarially perturbed by an *iterative* projected gradient update method; (II) whether the attack leverages *first-order gradient* of the model; (III) whether the $L_p$ boundary types are the same for the two stages, *i.e.*, attack and exploitation vectors; (IV) whether the loss functions for the two stages are the same; (V) whether the labels used (if any) for the two stages are relevant. Namely, only when the attack and exploitation vectors "match", SAE can be uniquely triggered as the exploitation vectors "continue" an attack, or they will "restart" an attack. Thus, in Fig. 1, Fig. 2 and Fig. 3, all the conditions are true as they involve PGD-like attacks. We acknowledge the ARC being insensitive to non-PGD-like attacks (such as C&W [12]) is a *limitation* in practice. However, the unique SAE meanwhile shows possibility of inferring the attack details mentioned in the above conditions once triggered. SAE is the trace of PGD-like attacks. Ablations for these five conditions are presented in Sec. 5.

**Adaptive Attack against ARC.** Adaptive attacks can be designed against defense [13] or detection [14]. Likewise, they can be designed against ARC feature. To avoid SAE in ARCm, the adaptive attack must reach a point where the corresponding ARCm has a mean value as small as that for benign examples. Intuitively, an adaptive attack has to simultaneously solve $\min_r \|\boldsymbol{S}_*(\boldsymbol{x}+\boldsymbol{r})\|_F$ (Frobenius norm) alongside its original attack goal. It however requires gradient of the Jacobians, namely at least $T+1$ Hessian matrices, *i.e.*, $\nabla^2 f_n(\cdot)$ of size $M \times M$ to perform gradient descent. This is computationally prohibitive as in the typical ImageNet setting (*i.e.*, $M{=}3{\times}224{\times}224$), a Hessian in `float32` precision needs $84.4$GiB memory. At this point, the cost of adaptive attack is much higher than computing ARC. We conclude that it is impractical to hide SAE from ARC at an acceptable cost without significant algorithm modification. The viable ways for attacker to avoid SAE is to use non-PGD-like attacks or break the SAE uniqueness conditions. Being resistant to adaptive attacks while surviving our extremely limited problem setting is left for future study.

## 3 Attack Detection and Inferring Attack Details

Attack detection aims to identify the attempt to adversarially perturb an image *even if* it fails to change the prediction (but meanwhile left the trace).[1] As demonstrated in the previous section, the SAE indicates the feasibility of attack detection specifically against PGD-like attacks.

**Informed Attack Detection** is to determine whether an arbitrary input $\tilde{\boldsymbol{x}}$ is adversarially perturbed, while the perturbation magnitude $\varepsilon$ is *known*. It can be viewed a binary classification problem, where the input is ARCv of $\tilde{\boldsymbol{x}}$, and the output $1$ indicates "adversarially perturbed", while $0$ indicates "unperturbed". Thus, for a given $\varepsilon = 2^k/255$ where $k \in \{1, 2, 3, 4\}$, a corresponding Support Vector Machine (SVM) [15] classifier $h_k(\tilde{\boldsymbol{x}}) \in \{0, 1\}$ can be trained using some benign ($\varepsilon{=}0$) samples and their adversarial counterparts ($\varepsilon{=}2^k/255$). Even if the training data only involves the BIM attack, from visualization results, we expect generalization for other PGD-like attacks despite domain shift.

**Uninformed Attack Detection** is to determine whether an arbitrary input $\tilde{\boldsymbol{x}}$ is adversarially perturbed, while the perturbation magnitude $\varepsilon$ is *unknown*. It can be viewed as an ordinal regression [16] problem, where the input is ARCv, and the output is the estimation of $k$, namely $\hat{k} \in \{0, 1, 2, 3, 4\}$. The corresponding estimate of $\varepsilon$ is $\hat{\varepsilon} = \mathbf{1}\{\hat{k} > 0\}2^{\hat{k}}/255$, where $\mathbf{1}\{\cdot\}$ is the indicator function.

---

[1] In practice it is undesirable to wait and react until the attack has succeeded.

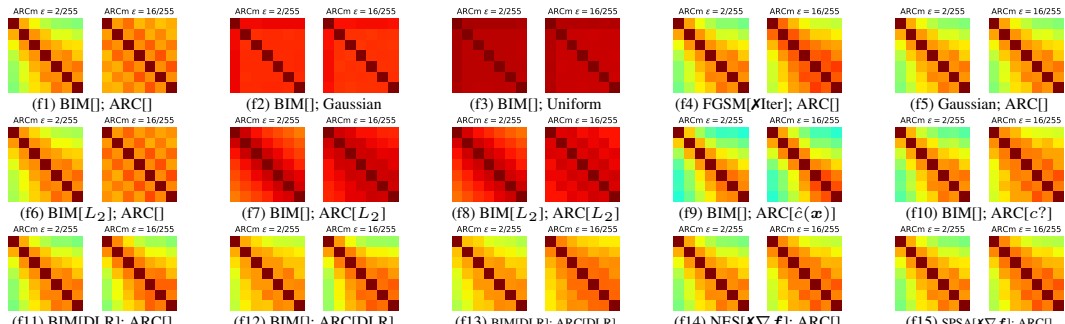

Figure 4: Ablation on SAE uniqueness by adjusting exploitation vectors for ARC. Each subfigure of ARCm pair has two annotations: (1) attack and its settings, where empty brackets means default setting unless overriden: $[L_p$ is $L_\infty$; Loss is $L_{CE}$; ✓(is) iterative; ✓(can access) gradient $\nabla f(\cdot)]$; (2) exploitation vector settings, *e.g.* "ARC[]" with the default setting $[L_p$ is $L_\infty$; Loss is $L_{CE}$; Label is $\check{c}(\cdot)]$. The "$c?$" means random guess. This figure is supplementary to Tab. 2.

Specifically, this is implemented as a series of binary classifiers (SVM), where the $k$-th ($k\neq0$) classifier predicts whether the level of perturbation is greater or equal to $k$, *i.e.*, whether $\hat{k} \geqslant k$. Note, based on our visualization, the ARCv cluster of adversarial examples is moving away from that of benign examples as $\varepsilon$ (or $k$) increases. This means the ARCv of an adversarial example with $\hat{k} \geqslant k$ will also cross the decision boundary of the $k$-th SVM $h_k(\cdot)$. Namely the SVM $h_k(\cdot)$ can also tell whether $\hat{k} \geqslant k$, and thus can be reused. Finally, the ordinal regression model can be expressed as the sum of prediction over the SVMs: $\hat{k} = \sum_{k\in\{1,2,3,4\}} h_k(\tilde{x})$. A perturbation is detected as long as $\hat{k} > 0$. Estimating $k$ (or $\varepsilon$) for $\tilde{x}$ is similar to matching its ARCm position inside a much larger ARCm calculated starting from benign example. But, the estimate does not have to be precise, because the detection is already successful once any of the SVMs correctly raises an alert.

Although a detector in practice knows completely nothing about a potential attack including the attack type, evaluation of uninformed attack detection with *known* attack type is enough. Regarding the performance for uninformed attack detection given a specific attack type of attack as a conditional performance, the expected performance in the wild can be calculated as the sum of conditional performance weighted by the prior probabilities that the corresponding attack happens.

**Inferring Attack Details.** Due to the SAE uniqueness in Sec. 2, once attack is detected, we can also predict that the attack: (I) is an iterative method performing projected gradient updates; (II) can access the first-order gradient of $f(\cdot)$; (III) uses the same type of $L_p$ bound as that in creation of exploitation vectors ($L_\infty$ by default); (IV) uses the same function as that in creation of exploitation vectors ($L_{CE}(\cdots)$ by default); (V) uses a ground-truth label which is relevant to the least-likely class $\check{c}(\tilde{x})$ used for exploitation vectors (in many cases $\check{c}(\tilde{x})$ is exactly the ground-truth). In other words, a feasible post-processing defense is to correct prediction into the least-likely class $\check{c}(\tilde{x})$ upon detection. Namely, the disadvantage of ARC being insensitive to non-PGD-like attacks is meanwhile advantage of being able to infer attack details of PGD-like attacks.

## 4 Experiments

In this section, we quantitatively verify the effectiveness of the ARC features in attack detection, and the performance of the post-processing defense under an *extremely limited setting*. Unlike related works, the MNIST evaluation is omitted, as the corresponding conclusions may not hold [14] on CIFAR-10, let alone ImageNet. We evaluate ResNet-18 [8] on CIFAR-10 [7]; ResNet-152 [8] and SwinT-B-IN1K [10] on ImageNet [9] with their official pre-trained weights (advantage of being non-intrusive). Our code is implemented based on PyTorch [17], TorchAttacks [18] and Foolbox [19].

**ARC Feature Parameter.** For the BIM attack for exploitation vectors, we set step number $T = 6$, and step size $\alpha = 2/255$ under the $L_\infty$ bound with $\varepsilon = 8/255$. Note, the mean value of ARCm will tend to 1 with a larger $T$, making ARCv less separable. We choose $T = 6$ to clearly visualize the value changes within ARCm, but this does not necessarily lead to the best performance.

Table 1: Informed and Uninformed (the "$\varepsilon$=?" column) Attack Detection. All numbers are percentage with the "%" sign omitted, except for MAE. Numbers greater than 50% are highlighted in bold font.

| Dataset Model | Attack | $\epsilon = 2/255$ | | | | $\epsilon = 4/255$ | | | | $\epsilon = 8/255$ | | | | $\epsilon = 16/255$ | | | | $\epsilon =?$ | | | | |
|---|---|---|---|---|---|---|---|---|---|---|---|---|---|---|---|---|---|---|---|---|---|---|
| | | DR | FPR | Acc | Acc* | DR | FPR | Acc | Acc* | DR | FPR | Acc | Acc* | DR | FPR | Acc | Acc* | MAE | DR | FPR | Acc | Acc* |
| CIFAR-10 ResNet-18 | BIM | 0.0 | 0.0 | 33.5 | 33.5 | 0.0 | 0.0 | 6.4 | 6.4 | 32.3 | 1.5 | 0.4 | 17.8 | **79.2** | 1.1 | 0.0 | **62.4** | 1.55 | 30.9 | 1.5 | 10.1 | 30.7 |
| | PGD | 0.0 | 0.0 | 33.7 | 33.7 | 0.0 | 0.0 | 6.4 | 6.4 | 33.0 | 1.5 | 0.4 | 18.6 | **81.2** | 1.1 | 0.0 | **64.8** | 1.54 | 31.5 | 1.5 | 10.1 | 31.5 |
| | MIM | 0.0 | 0.0 | 30.4 | 30.4 | 0.0 | 0.0 | 6.5 | 6.5 | 37.5 | 1.5 | 0.4 | 22.3 | **84.5** | 1.1 | 0.0 | **67.4** | 1.50 | 33.6 | 1.5 | 9.3 | 32.4 |
| | APGD | 0.0 | 0.0 | 29.3 | 29.3 | 0.0 | 0.0 | 5.1 | 5.1 | 36.9 | 1.5 | 0.2 | 20.7 | **78.8** | 1.1 | 0.0 | **55.8** | 1.53 | 31.5 | 1.5 | 8.7 | 28.0 |
| | AA | 0.0 | 0.0 | 27.4 | 27.4 | 0.0 | 0.0 | 2.1 | 2.1 | 37.3 | 1.5 | 0.1 | 20.6 | **78.4** | 1.1 | 0.0 | **55.6** | 1.53 | 31.6 | 1.5 | 7.4 | 26.8 |
| | ? | 0.0 | 0.0 | 30.9 | 30.9 | 0.0 | 0.0 | 5.3 | 5.3 | 35.4 | 1.5 | 0.3 | 20.0 | **80.4** | 1.1 | 0.0 | **61.2** | 1.53 | 31.8 | 1.5 | 9.1 | 29.9 |
| ImageNet ResNet-152 | BIM | 0.0 | 0.0 | 0.0 | 0.0 | 4.7 | 1.4 | 0.0 | 0.0 | 20.5 | 1.4 | 0.0 | 0.0 | **91.6** | 1.4 | 0.0 | 0.4 | 1.36 | 30.6 | 1.6 | 0.0 | 0.1 |
| | PGD | 0.0 | 0.0 | 0.0 | 0.0 | 4.7 | 1.4 | 0.0 | 0.0 | 18.8 | 1.4 | 0.0 | 0.0 | **85.9** | 1.4 | 0.0 | 0.0 | 1.44 | 28.9 | 1.6 | 0.0 | 0.0 |
| | MIM | 0.0 | 0.0 | 0.0 | 0.0 | 2.3 | 1.4 | 0.0 | 0.0 | 4.7 | 1.4 | 0.0 | 0.0 | **81.2** | 1.4 | 0.0 | 0.0 | 1.52 | 23.8 | 1.6 | 0.0 | 0.2 |
| | APGD | 0.0 | 0.0 | 0.0 | 0.0 | 2.0 | 1.4 | 0.0 | 0.0 | 11.3 | 1.4 | 0.0 | 0.0 | **61.7** | 1.4 | 0.0 | 0.4 | 1.59 | 19.7 | 1.6 | 0.0 | 0.1 |
| | AA | 0.0 | 0.0 | 0.0 | 0.0 | 2.5 | 1.4 | 0.0 | 0.0 | 10.7 | 1.4 | 0.0 | 0.0 | **61.5** | 1.4 | 0.0 | 0.0 | 1.59 | 19.9 | 1.6 | 0.0 | 0.0 |
| | ? | 0.0 | 0.0 | 0.0 | 0.0 | 3.2 | 1.4 | 0.0 | 0.0 | 13.2 | 1.4 | 0.0 | 0.0 | **76.3** | 1.4 | 0.0 | 0.2 | 1.50 | 24.6 | 1.6 | 0.0 | 0.1 |
| ImageNet SwinT-B-IN1K | BIM | 4.1 | 1.6 | 6.1 | 6.2 | 13.7 | 2.0 | 0.0 | 8.4 | **77.3** | 2.0 | 0.0 | **74.0** | **97.9** | 0.2 | 0.0 | **97.9** | 0.96 | 49.1 | 2.0 | 1.5 | 47.3 |
| | PGD | 3.9 | 1.6 | 2.3 | 3.1 | 16.4 | 2.0 | 0.0 | 10.9 | **72.7** | 2.0 | 0.0 | **68.8** | **98.4** | 0.2 | 0.0 | **98.4** | 1.01 | 48.6 | 2.0 | 0.6 | 45.9 |
| | MIM | 1.6 | 1.6 | 0.0 | 1.6 | 10.2 | 2.0 | 0.0 | 10.2 | **63.3** | 2.0 | 0.0 | **63.3** | **93.8** | 0.2 | 0.0 | **93.8** | 1.09 | 43.8 | 2.0 | 0.0 | 43.8 |
| | APGD | 1.4 | 1.6 | 0.0 | 1.0 | 5.3 | 2.0 | 0.0 | 4.5 | 32.6 | 2.0 | 0.0 | 25.2 | **65.0** | 0.2 | 0.0 | **51.0** | 1.37 | 29.4 | 2.0 | 0.0 | 23.2 |
| | AA | 1.8 | 1.6 | 0.0 | 1.0 | 5.7 | 2.0 | 0.0 | 4.3 | 31.6 | 2.0 | 0.0 | 25.0 | **68.4** | 0.2 | 0.0 | **54.1** | 1.37 | 29.5 | 2.0 | 0.0 | 23.2 |
| | ? | 2.6 | 1.6 | 1.7 | 2.6 | 10.2 | 2.0 | 0.0 | 7.7 | **55.5** | 2.0 | 0.0 | **51.2** | **84.7** | 0.2 | 0.0 | **79.0** | 1.16 | 40.1 | 2.0 | 0.4 | 36.7 |

**Training.** To train SVMs $h_k(\cdot)$ with RBF kernel, we randomly select **50** training samples from CIFAR-10, and perturb them using *only* BIM [1] with magnitude $\varepsilon = 2/255, 4/255, 8/255, 16/255$, respectively. Then each of the four $h_k(\cdot)$ is trained with ARCv of the benign ($\varepsilon = 0$) samples and perturbed ($\varepsilon = 2^k/255$) samples. Likewise, for ImageNet we randomly select **50** training samples and train SVM in a similar setting separately for ResNet-152 and SwinT-B-IN1K. The weight for benign sample can be adjusted for training in order to control False Positive Rate (FPR).

**Testing.** For CIFAR-10, all 10000 testing data and their perturbed versions with different $\varepsilon$ are used to test our SVM. For ImageNet, we randomly choose 512 testing samples to test our SVM due to computation cost of Jacobian matrices. A wide range of adversarial attacks are involved, including (1) PGD-like attacks: include BIM [1], PGD [2], MIM [3], APGD [4], AutoAttack (AA) [4]; (2) Non-PGD-like attacks: (2.1) other white-box attacks: FGSM [6], C&W [12] (we use $\varepsilon \in \{0.5, 1.0, 2.0, 3.0\}$ in $L_2$ case), FAB [20], FMN [21]; (2.2) transferability-based attacks: DI-FGSM [22], TI-FGSM [23] (using ResNet-50 as proxy); (2.3) score-based black-box methods: NES [24], SPSA [25], Square [26]. Existing attack detection methods seldom evaluate on many types of attacks. AutoAttack is regarded as PGD-like because APGD is its most significant component for attack success rate. Details of all attacks can be found in the supplementary code.

**Metrics.** We evaluate the SVMs using Detection Rate (DR, *a.k.a.*, True Positive Rate), as well as False Positive Rate (FPR). For the post-processing defense method, we report the original classification accuracy for perturbed examples (denoted as "Acc") as well as accuracy after correction (denoted as "Acc*"). For ordinal regression, we also report Mean Average Error (MAE) for reference.

## 4.1 Informed and Uninformed Attack Detection for PGD-like Attacks

For each network, the corresponding SVMs are trained and evaluated as shown in Tab. 1. Columns with a concrete $\varepsilon$ value are informed attack detection, while the "$\varepsilon$=?" column is un-informed attack detection. As can be expected from visualization results, the ARCv clusters are gradually becoming separatable with $\varepsilon$ increasing, and hence the increase of DR. Notably, the large perturbations (*i.e.*, $\varepsilon = 16/255$) are very hard to defend [27], but can be consistently and accurately detected across architectures. The ARC feature is especially effective for Swin-Transformer, because this model transitions faster from being non-linear to being linear than other architectures. Such characteristics are beneficial for ARC.

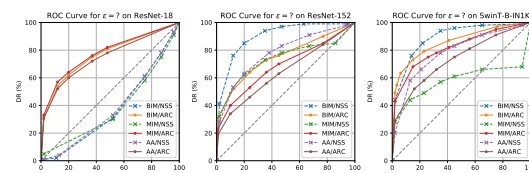

Figure 5: ROC of SVMs in Tab. 1 & Tab. 3.

Upon detection of attack, our method corrects the prediction into the least-likely class as a post-processing defense. Success of such method depends on whether the attack is efficient to make ground-truth class least-likely, and whether the network is easy for the attack to make a class least-likely. From Tab. 1, both ResNet-18 and SwinTransformer have such property and lead to high

Table 2: Ablation on SAE uniquenss by varying attacks. The row (t1) is regarded as a baseline, and notation ".." means "same as baseline" in order to ease comparison. SAE will only show consistent effectiveness across architectures when the four conditions in Sec. 2 are satisfied.

| # | Attack | | | | | ARC | | | ResNet-18 w/ $\epsilon$ =? | | | | | ResNet-152 w/ $\epsilon$ =? | | | | | SwinT-B-IN1K w/ $\epsilon$ =? | | | | |
|---|---|---|---|---|---|---|---|---|---|---|---|---|---|---|---|---|---|---|---|---|---|---|---|
| | Name | $L_p$ | Loss | Iter. | $\nabla f(\cdot)$ | $L_p$ | Loss | Label | MAE | DR | FPR | Acc | Acc* | MAE | DR | FPR | Acc | Acc* | MAE | DR | FPR | Acc | Acc* |
| t1 | BIM | $\infty$ | CE | Yes | Yes | $\infty$ | CE | $\check{c}(\boldsymbol{x})$ | 1.55 | 30.9 | 1.5 | 10.1 | 30.7 | 1.36 | 30.6 | 1.6 | 0.0 | 0.1 | 0.96 | 49.1 | 2.0 | 1.5 | 47.3 |
| t2 | BIM | 2 | .. | .. | .. | .. | .. | .. | 1.27 | 49.9 | 1.5 | 2.6 | 39.0 | 1.98 | 3.5 | 1.6 | 0.2 | 0.2 | 2.02 | 1.0 | 2.0 | 1.4 | 1.8 |
| t3 | BIM | .. | DLR | .. | .. | .. | .. | .. | 1.98 | 2.1 | 1.5 | 10.5 | 10.6 | 1.63 | 18.9 | 1.6 | 0.0 | 0.6 | 1.44 | 27.5 | 2.0 | 1.8 | 6.6 |
| t4 | FGSM | .. | .. | No | .. | .. | .. | .. | 1.96 | 3.4 | 1.5 | 30.3 | 29.5 | 1.63 | 18.6 | 1.6 | 8.4 | 6.8 | 1.44 | 27.1 | 2.0 | 44.9 | 32.4 |
| t5 | C&W | 2 | C&W | .. | .. | .. | .. | .. | 1.99 | 1.2 | 1.5 | 0.0 | 0.0 | 2.02 | 2.3 | 1.6 | 0.0 | 0.0 | 2.03 | 1.6 | 2.0 | 0.0 | 0.0 |
| t6 | FAB | .. | FAB | .. | .. | .. | .. | .. | 1.99 | 1.0 | 1.5 | 10.6 | 10.5 | 2.00 | 2.5 | 1.6 | 9.2 | 9.2 | 2.03 | 0.8 | 2.0 | 9.4 | 9.4 |
| t7 | FMN | .. | FMN | .. | .. | .. | .. | .. | 1.99 | 1.4 | 1.5 | 8.8 | 8.6 | 2.02 | 2.1 | 1.6 | 0.0 | 0.0 | 2.03 | 0.8 | 2.0 | 0.0 | 0.0 |
| t8 | DI-FGSM | .. | DI-FGSM | .. | No | .. | .. | .. | 1.98 | 2.2 | 1.5 | 42.9 | 42.0 | 1.98 | 3.5 | 1.6 | 27.9 | 27.5 | 1.87 | 8.2 | 2.0 | 67.2 | 62.1 |
| t9 | TI-FGSM | .. | TI-FGSM | .. | No | .. | .. | .. | 1.98 | 1.9 | 1.5 | 59.4 | 58.3 | 2.00 | 2.9 | 1.6 | 40.0 | 39.1 | 2.02 | 1.6 | 2.0 | 72.3 | 70.9 |
| t10 | NES | .. | .. | .. | No | .. | .. | .. | 1.94 | 4.7 | 1.5 | 38.6 | 39.4 | 1.98 | 3.1 | 1.6 | 28.3 | 27.3 | 2.02 | 1.6 | 2.0 | 50.6 | 49.4 |
| t11 | SPSA | .. | .. | .. | No | .. | .. | .. | 1.97 | 3.0 | 1.5 | 39.2 | 39.1 | 2.00 | 3.1 | 1.6 | 29.9 | 28.9 | 2.00 | 2.7 | 2.0 | 52.7 | 50.6 |
| t12 | Square | .. | Square | .. | No | .. | .. | .. | 1.99 | 1.6 | 1.5 | 85.7 | 84.3 | 2.02 | 2.1 | 1.6 | 68.6 | 67.4 | 1.84 | 10.2 | 2.0 | 77.9 | 70.1 |
| t13 | Gaussian | .. | N/A | No | No | .. | .. | .. | 1.99 | 1.7 | 1.5 | 87.0 | 85.6 | 2.00 | 2.7 | 1.6 | 75.2 | 73.2 | 2.00 | 3.1 | 2.0 | 82.4 | 79.7 |
| t14 | Uniform | .. | N/A | No | No | .. | .. | .. | 1.99 | 1.8 | 1.5 | 86.6 | 85.0 | 1.97 | 4.1 | 1.6 | 73.6 | 70.9 | 1.84 | 10.2 | 2.0 | 81.8 | 73.2 |

classification accuracy after correction. For ResNet-152, the least-likely label is merely relevant (not identical) to the ground-truth due to network property during attack, and hence leads to effective detection but not correction (this will be explained in next subsection). In contrast, the correction method performs best on Swin-Transformer, as it can restore classification accuracy from $0.4\%$ to $36.7\%$ even if both concrete type of PGD-like attack and $\varepsilon$ are unknown ("Attack=?" row and "$\varepsilon$=?" column in Tab. 1), assuming flat prior. By adjusting the weights assigned to benign examples, the decision boundary of SVMs can be moved and hence influence the FPR, as shown in in Fig. 5.

## 4.2 Sequel Attack Effect as Unique Trace of PGD-like Attacks

The SAE is unique to PGD-like attacks, as it requires five conditions listed in Sec. 2 to hold for consistent effectiveness. To clarify this, we change the attack settings (quantitatively in Tab. 2), or the exploitation vector for ARCm (qualitatively on CIFAR10 in Fig. 4), and then review these conditions:

(I). Iterative attack (Iter.). The single-step version of PGD, *i.e.*, FGSM (t4, f4) does not effectively exploit the search space within the $L_p$ bound, and hence will not easily trigger linearity and SAE. Only Swin Transformer slightly reacts against FGSM due to its own characteristics of being easy to be turned linear. Thus, SAE requires the attack to be iterative;

(II). Gradient access ($\nabla f(\cdot)$). Transferability-based attacks (t8, t9) uses proxy model gradients to create adversarial examples, and hence could not trigger SAE. NES (t10, f14) and SPSA (t11, f15) can be seen as PGD using gradients estimated from only network logits, but can still not trigger SAE as it cannot efficiently trigger linearity. Neither does Square attack (t12). Thus, SAE requires that the attacks use the target model gradient;

(III). Same $L_p$ bound. When the attack is BIM in $L_2$ bound (t2, f6), SAE will no longer be triggered for ImageNet models, because the change of $L_p$ influences perturbation search process. However, SAE is still triggered for CIFAR-10 possibly due to relatively low-dimensional search space. This means CIFAR-10 property does not necessarily generalize to ImageNet. When ARC is changed accordingly (f7, f8), the feature clusters are still separatable. Thus, SAE requires the same type of $L_p$ bound for consistent effectiveness;

(IV). Same loss. When the loss for the BIM attack is switched from $L_{CE}$ to DLR [4] (t3, f11), the SAE is significantly reduced. However, if exploitation vectors are also created using DLR loss (f12, f13), SAE will be triggered again. Thus, SAE requires a consistent loss function;

(V). Relevant label. When the most-likely label $\hat{c}(\tilde{\boldsymbol{x}})$ is used for exploitation vectors, it leads to the least significant SAE (f9). Besides, even a random label ($c$?) leads to moderate SAE (f10), while the least-likely label $\check{c}(\tilde{\boldsymbol{x}})$ (which is ground-truth label in many cases) leads to distinct SAE (f1). The most significant SAE correspond to $\check{c}(\tilde{\boldsymbol{x}}) = c(\boldsymbol{x})$. This means in order to maximize cross-entropy, a large portion of output functions $f_n(\cdot)$ has been triggered local linearity during attack. Thus, SAE requires a relevant label (if any) for exploitation vectors.

When the exploitation vectors are created using random noise (f2, f3), SAE is not triggered. Neither does random noise as attack trigger SAE (t13, t14, f5). Other non-PGD-like attacks (t5, t6, t7) do not trigger SAE as well. A special case is targeted PGD-like attack, where the creation of exploitation vector needs to be use negative cross-entropy loss on the most-likely label to reach a similar level of effectiveness (this paper focuses on the default untargeted attack to avoid complication).

Table 3: Comparison with existing methods that are compatible with our problem setting.

| Method | Metric | BIM | | | | | PGD | | | | | MIM | | | | | APGD | | | | | AA | | | | |
|---|---|---|---|---|---|---|---|---|---|---|---|---|---|---|---|---|---|---|---|---|---|---|---|---|---|---|
| | | 2/255 | 4/255 | 8/255 | 16/255 | ? | 2/255 | 4/255 | 8/255 | 16/255 | ? | 2/255 | 4/255 | 8/255 | 16/255 | ? | 2/255 | 4/255 | 8/255 | 16/255 | ? | 2/255 | 4/255 | 8/255 | 16/255 | ? |
| **CIFAR10 ResNet-18** | | | | | | | | | | | | | | | | | | | | | | | | | | |
| NSS [29] DR | DR | 0.0 | 0.0 | 0.0 | 0.1 | 0.5 | 0.0 | 0.0 | 0.0 | 0.1 | 0.5 | 0.0 | 0.0 | 0.0 | 0.1 | 4.7 | 0.0 | 0.0 | 0.3 | 0.2 | 0.8 | 0.0 | 0.0 | 0.3 | 0.2 | 0.8 |
| | FPR | 0.0 | 0.0 | 1.8 | 1.5 | 2.5 | 0.0 | 0.0 | 1.8 | 1.5 | 2.5 | 0.0 | 0.0 | 1.8 | 1.5 | 2.5 | 0.0 | 0.0 | 1.8 | 1.5 | 2.5 | 0.0 | 0.0 | 1.8 | 1.5 | 2.5 |
| ARC | DR | 0.0 | 0.0 | 32.3 | 79.2 | 30.9 | 0.0 | 0.0 | 33.0 | 81.2 | 31.5 | 0.0 | 0.0 | 37.5 | 84.5 | 33.6 | 0.0 | 0.0 | 36.9 | 78.8 | 31.5 | 0.0 | 0.0 | 37.3 | 78.4 | 31.6 |
| | FPR | 0.0 | 0.0 | 1.5 | 1.1 | 1.5 | 0.0 | 0.0 | 1.5 | 1.1 | 1.5 | 0.0 | 0.0 | 1.5 | 1.1 | 1.5 | 0.0 | 0.0 | 1.5 | 1.1 | 1.5 | 0.0 | 0.0 | 1.5 | 1.1 | 1.5 |
| **ImageNet ResNet-152** | | | | | | | | | | | | | | | | | | | | | | | | | | |
| NSS [29] | DR | 2.9 | 19.1 | 39.6 | 47.2 | 41.6 | 2.9 | 19.9 | 39.6 | 46.5 | 41.1 | 4.2 | 31.2 | 41.4 | 9.1 | 32.9 | 1.1 | 12.6 | 28.3 | 35.7 | 29.1 | 1.0 | 11.9 | 29.8 | 33.3 | 28.7 |
| | FPR | 0.4 | 1.4 | 1.2 | 1.4 | 2.0 | 0.4 | 1.4 | 1.2 | 1.4 | 2.0 | 0.4 | 1.4 | 1.2 | 1.4 | 2.0 | 0.6 | 1.4 | 1.2 | 1.4 | 2.0 | 0.4 | 1.4 | 1.2 | 1.4 | 2.0 |
| ARC | DR | 0.0 | 4.7 | 20.5 | 91.6 | 30.6 | 0.0 | 4.7 | 18.8 | 85.9 | 28.9 | 0.0 | 2.3 | 4.7 | 81.2 | 23.8 | 0.0 | 2.0 | 11.3 | 61.7 | 19.7 | 0.0 | 2.5 | 10.7 | 61.5 | 19.9 |
| | FPR | 0.0 | 1.4 | 1.4 | 1.4 | 1.6 | 0.0 | 1.4 | 1.4 | 1.4 | 1.6 | 0.0 | 1.4 | 1.4 | 1.4 | 1.6 | 0.0 | 1.4 | 1.4 | 1.4 | 1.6 | 0.0 | 1.4 | 1.4 | 1.4 | 1.6 |
| **ImageNet SwinT-B-IN1K** | | | | | | | | | | | | | | | | | | | | | | | | | | |
| NSS [29] | DR | 4.5 | 16.2 | 42.4 | 47.5 | 44.2 | 4.9 | 15.8 | 41.8 | 47.1 | 44.1 | 12.3 | 28.7 | 29.3 | 4.5 | 28.9 | 1.6 | 11.0 | 31.3 | 35.5 | 31.1 | 1.4 | 10.4 | 31.8 | 35.1 | 30.8 |
| | FPR | 0.6 | 1.0 | 1.2 | 1.6 | 2.3 | 0.6 | 1.0 | 1.2 | 1.6 | 2.3 | 0.6 | 1.0 | 1.2 | 1.5 | 2.3 | 0.6 | 1.0 | 1.2 | 1.6 | 2.3 | 0.6 | 1.0 | 1.2 | 1.6 | 2.3 |
| ARC | DR | 4.1 | 13.7 | 77.3 | 97.9 | 49.1 | 3.9 | 16.4 | 72.7 | 98.4 | 48.6 | 1.6 | 10.2 | 63.3 | 93.8 | 43.8 | 1.4 | 5.3 | 32.6 | 65.0 | 29.4 | 1.8 | 5.7 | 31.6 | 68.4 | 29.5 |
| | FPR | 1.6 | 2.0 | 2.0 | 0.2 | 2.0 | 1.6 | 2.0 | 2.0 | 0.2 | 2.0 | 1.6 | 2.0 | 2.0 | 0.2 | 2.0 | 1.6 | 2.0 | 2.0 | 0.2 | 2.0 | 1.6 | 2.0 | 2.0 | 0.2 | 2.0 |

The non-PGD attacks, or PGD attacks do not meed all conditions cannot consistently trigger SAE across architectures because they provide a less "matching" starting point for exploitation vectors, and hence make the BIM for exploitation vectors "restart" an attack, where the network behaves non-linear again. Only when all the conditions are satisfied will SAE be consistently triggered across different architectures, especially for ImageNet models. As for label correction, PGD-like attacks can effectively leak the ground-truth labels in the adversarial example, as long as the network allows the attack to easily reduce the corresponding logit value to lowest among all.

In summary, SAE is the unique trace of PGD-like attacks. Although insensitive to non-PGD-like attacks for general attack detection, SAE is a specific signature [28], indicating the feasibility of correcting prediction upon detection of PGD-like attacks.

### 4.3 Comparison with Previous Attack Detection Methods

As discussed in Sec. 6, due to our extremely limited problem setting – (1) no auxiliary deep model; (2) non-intrusive; (3) data-undemanding, the most relevant methods that do not lack of ImageNet evaluation are [29, 30, 31, 32, 33]. But [30, 31, 32, 33] still require a considerable amount of data to build accurate (relatively) high-dimensional statistics. The remaining NSS [29] method craft 18-dimensional features from Natural Scene Statistics, which are fed into SVM for binary classification. We adapt the trained SVMs in our ordinal regression framework as well, with a reduced training set size to 100 (50 benign + 50 BIM adversarial) for each SVM for fair comparison. All SVMs are tuned to control FPR. The results and ROC curves for "$\varepsilon{=}?$" task can be found Tab. 3 and Fig. 5. It is noted that (1) SVM with the 18-D NSS feature may fail to generalize due to insufficient sampling (hence the below-diagonal ROC); (2) NSS performs better for small $\varepsilon$, but performance saturates with larger $\varepsilon$, because NSS does not incorporate any cue from network gradient behavior; (3) small $\varepsilon$ is difficult for ARC, but its performance soars with larger $\varepsilon$ towards $100\%$, which is consistent and expected from our visualization; (4) SVM with ARCv can generalize against all PGD-like attacks, while NSS failed for MIM; (5) SVM with NSS may generalize against some non-PGD-like attacks [29], while ARC could not due to SAE uniqueness; (6) SVM with the 2-D NSS feature ("Method 2" in [29]) fails to generalize. Thus, ARC achieves competitive performance consistently across different settings despite the extreme limits, because the ARC feature is low-dimensional, and incorporates cue from network gradient behavior. Apart from these, ARC also provides a new perspective to understanding attack and defense from model's gradient behavior, as discussed in Sec. 5.

## 5 Discussions and Justifications

**Ordinal Regression.** Intuitively, the uninformed attack detection can be formulated as standard regression to estimate a continuous $k$ value. However, this introduces an undesired additional threshold hyper-parameter for deciding whether an input with *e.g.*, 0.5 estimation is adversarial. Ordinal regression produces discrete $k$ values and avoids such ambiguity and unnecessary parameter.

**Training Set Size.** Each of our SVMs has only 100 training data (*i.e.*, 50 benign + 50 adversarial). The simple 2-D ARCv distribution (Fig. 2) can be reflected by few data points, which even allows an SVM to generalize with less than 100 data points (but may suffer from insufficient sampling with too few, *e.g.*, 10+10 samples). In contrast, the performance gain will be marginal starting from roughly 200 training samples, because the ARCv feature distribution is already well represented.

**Combination with Adversarial Training.** From our experiment and recent defenses [2, 34, 27], its noted that (1) small perturbations are hard to detect, but easy to defend; while (2) large perturbations are hard to defend, but easy to detect. However, combining defense and our detection is not effective on ImageNet. As shown in Fig. 6, we compute ARCm based on regular ResNet-50 (from PyTorch [17]) and adversarially trained ResNet-50 on ImageNet (from [34]). Unlike the regular ResNet-50, adversarially trained one has much higher mean value in ARCm, and the resulting ARC vectors are almost non-separatable. This means adversarial training makes the model very linear around the data [35]. As a new perspective on why adversarial training works, the networks are trained to generalize while being already very linear to the input, and thus it will be hard for attack to make the model behave even more linear to significantly manipulate the output.

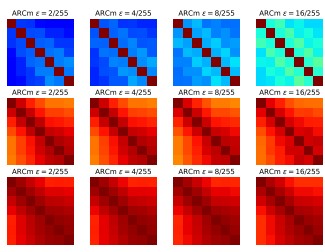

Figure 6: ARCm from regular (1st row), and adversarially trained ResNet-50 (2nd row w/ $\varepsilon=4/255$, 3rd row w/ $\varepsilon=8/255$).

**Limitations.** (1) The ARC Feature is only sensitive to the PGD-like attacks, and relies on the least-likely assumption for effectiveness of prediction correction. But such selective sensitivity meanwhile leads to the uniqueness of SAE. (2) Jacobian computation is slow for ImageNet models because it requires 1000 iterations of backward pass. A single Jacobian of ResNet-152 takes $161\pm0.5$ seconds on Nvidia Titan Xp. Thus we are unable evaluate our method on all ImageNet data with 2 GPUs.

**Future Recommendations.** (1) Include ImageNet evaluation, as CIFAR-10 property may not hold on ImageNet; (2) Check detector sensitivity *w.r.t.* attack algorithm parameter, as it may be significant.

## 6   Related Works

**Adversarial Attack and Defense.** Neural networks are vulnerable to attacks [36, 6, 12]. To exploit such vulnerability, attacks under different threat models are designed, including but not limited to white-box attacks [1, 2, 3, 4], transferability-based attacks [37, 38, 22, 23], score-based black-box attacks [39, 24, 25, 26], and decision-based black-box attacks [40]. Different from these run-time attacks, backdoor attack [41] happens during the training. To counter the attacks, adversarial training [2, 27, 42] is the most promising defense to make networks resistant to the adversarial perturbations, but is meanwhile intrusive (*i.e.* requires retraining), and suffering from a notable generalization gap. Certified defense [43] and perturbation reverse engineering are also proposed [44]. A defense may be invalidated by adaptive attacks [45, 13]. Our method to correct the prediction upon detection can be seen as a post-processing defense.

**Adversarial Example Detection** [46, 14] aims to predict whether a given image is adversarial or not, so that adversarial ones can be rejected. This can be achieved through adversarial training [47, 48], customized subnet [49] or customized loss [50], but will be costly for ImageNet. Generative model-based detection methods check adversarial example reconstruction error [51] or probability density [52], but are data-demanding in order to learn accurate distributions. Auxiliary deep model [53, 54] for attack detection not only require large amount of data, but are also susceptible to adaptive attack [14]. Dropout can be used for detection when combined with Bayesian uncertainty [55]. Feature statistics-based methods [31, 30, 29, 32, 33] leverage (high-dimensional) features, which is the most compatible group of method to our problem setting, but most of them are still data-demanding for an accurate statistics. Whilst MNIST property may not hold on CIFAR-10 [14], let alone ImageNet, many related works lack the evaluation on ImageNet. Whilst detection difficulty varies with attack parameters, a very large portion of related works have neglected the respective sensitivity analysis. Additionally, we point out conditions under which our method will be invalidated.

## 7   Conclusions

In this paper, we design an Adversarial Response Characteristic (ARC) feature with an intuition that the model being attacked behaves more "linear" against adversarial examples than does to benign ones, which is valid for PGD-like attacks in terms of attack detection and prediction correction. Our method is light-weighted, non-intrusive, data-undemanding and simple to interpret.

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

# A  Additional Discussions

## A.1  Summary of Pros & Cons of the Proposed Method

**Pros:**

- Relies on strong assumptions and hence is specifically effective for PGD-like attacks. Namely, the unique trace of PGD-like attacks can be used in specific (instead of generic) defense scenarios with knowledge about the attacker, or forensics scenarios to tell whether an adversarial example is created by PGD-like methods by identifying the unique trace.

- Can infer other attack algorithm details such as loss function and the ground-truth labels, while the other attack detection methods cannot do the same.

- Easy and straightforward to interpret for human, since the meaning of the ARC features is clearly defined, and the feature dimensionality is low.

- Light-weighted in terms of algorithm components. No any additional deep neural networks is required.

- Non-intrusive. Does not require any change in neural network architecture or parameters. The proposed method analyzes the Jacobian matrices calculated from the neural network of interest.

- Data-undemanding. Does not require a large number of training data. We use merely 50 training samples in our experiments.

- The stronger the attack is, the stronger the trace is (and hence the higher detection rate). Previous methods compatible to our extremely-limited setting do not have such property and may even perform worse with large perturbations in some cases (See Table 3).

- Reveals a new perspective to understand why Adversarial Training works. (See "Combination with Adversarial Training" in Section 5).

**Cons:**

- Relies on strong assumptions (See "Uniqueness of SAE to PGD-Like Attack" in Section 2), and hence is not effective under non-PGD scenarios since assumptions are broken. Ablation studies are carefully carried out in Section 4.2 to examine and justify these assumptions.

- Suffers from high time complexity due to Jacobian matrix calculation. In practice, this is reflected by time consumption of calculation of the ARC feature (See "Limitations" in Section 5). Experiments on ImageNet are extremely slow and hence we are unable to evaluate the method on all ImageNet data.

- Performs worse than previous NSS method against small perturbations (*i.e.*, $\varepsilon = 2/255$ or $\varepsilon = 4/255$). (But significantly better against large perturbations).

- Incompatible with Adversarial Training. (But meanwhile provides a new perspective to understand why adversarial training works. See "Combination with Adversarial Training" in Section 5).

## A.2  Iterations of PGD-like Attacks

It is known that the number of iterations (fixed at 100 in our experiments) also impacts the attack strength besides perturbation magnitude $\varepsilon$. As increasing number of iterations will also lead to a more linear response from the model given an fixed and appropriate $\varepsilon$ and achieve SAE similarly, we stick to one controlled variable $\varepsilon$ for simplicity.

On the contrary, reducing the number of iterations of a PGD-like attack will also lead to small perturbations that are hard to detect (as demonstrated in Section 4), and hence increase the possibility that the attack will not trigger clear SAE and hence bypass the proposed detection method. As an extreme case, FGSM, namely the single-step version of PGD does not effectively trigger SAE (as discussed in Section 4.2).

The related works usually fix at a single set of attack parameters, and hence miss the observation that smaller perturbations are harder to detect.

## A.3 Motivation of Extremely Limited Setting, including Limited Data

An extremely limited problem setting (Paragraph 1 in Section 1) makes the proposed method flexible and applicable in a wider range of defense and forensics scenarios compared to existing methods. Namely, a method can be used in more flexible scenarios if it requires less from the adopter.

**Limited number of data samples.** Data-demanding methods is only applicable for models using publicly available datasets, or is only applicable by the first-party who trained the neural network. This limits the use cases of these methods. In contrast, we do not assume collecting a large amount of data is easy for potential adopters of the proposed method. Due to the low demand on data, the proposed method enables a wider range of defense or forensics scenarios, especially when there is no access to the whole training dataset. For instance, the "Third-party Attack Detection or Forensics" and "Attack Detection for Federated Learning" scenarios.

- Third-party Attack Detection (identify whether the model is attacked) or Forensics (identify attack type and infer the attack detail). Being data-undemanding means the proposed method can be applied to any pre-trained neural network randomly downloaded from the internet, or purchased from an commercial entity. For pre-trained neural networks using proprietary training datasets with commercial secret or ethic/privacy concerns (such as commercial face datasets and CT scans from patients), the proposed method is still valid as long as there are are a few training samples for reference, or it is possible to request a few reference training samples.

- Attack Detection for Federated Learning. In federated learning, raw training data (such as face images) is forbidden to be transmitted to the central server. And hence even the neural network trainer cannot access the full training dataset (will violate user privacy), and it is impossible to use any data-demanding methods to detect attack against a trained model (*e.g.*, face recognition model). In contrast, the proposed method is still valid in this scenario as long as a few training samples can be collected from several volunteers for reference.

**No change to network architecture or weights.** Many models deployed in production are unaware of adversarial attack. Re-training and replacing these models will induce cost, and will even introduce the risk of reducing benign example performance.

**No auxiliary deep networks.** Since a large amount of data is assumed to be not easy to obtain due to commercial or ethic reasons, training auxiliary deep networks are not always feasible. Pre-trained auxiliary deep networks are not always available for any classification task.

## A.4 More on Adaptive Attack

According to [13], some similar attack detection methods are broken by adaptive attacks. Here we discuss more about the existing adaptive attacks and report the quantitative experimental results. We also further elaborate on the adaptive attack mentioned in Section 2.

**Logit Matching.** (from Section 5.2 "The Odds are Odd" of [13]) Instead of maximizing the default entropy loss, we switch to minimize the MSE loss between the clean logits from another class and that of the adversarial example. We conduct experiment with all testing data from CIFAR-10, and 128 random testing samples from ImageNet (due to limited time frame of rebuttal). The experimental results can be found in the following table. Note, switching loss function to MSE loss (Logit Matching) breaks our assumption (IV). However, the attack still triggers SAE through the least-likely class, and hence our method is still effective, but is (expectedly) weaker compared to the BIM with the original cross-entropy loss.

| Dataset Model | Attack | $\epsilon = 2/255$ | | | | $\epsilon = 4/255$ | | | | $\epsilon = 8/255$ | | | | $\epsilon = 16/255$ | | | | $\epsilon =?$ | | | |
|---|---|---|---|---|---|---|---|---|---|---|---|---|---|---|---|---|---|---|---|---|---|
| | | DR | FPR | Acc | Acc* | DR | FPR | Acc | Acc* | DR | FPR | Acc | Acc* | DR | FPR | Acc | Acc* | DR | FPR | Acc | Acc* |
| CIFAR-10 ResNet-18 | BIM (Logit Matching) | 0.0 | 0.0 | 80.6 | 80.6 | 0.0 | 0.0 | 63.2 | 63.2 | 23.8 | 1.5 | 46.3 | 35.5 | 48.0 | 1.1 | 38.0 | 20.2 | 22.8 | 1.5 | 57.1 | 46.9 |
| ImageNet ResNet-152 | BIM (Logit Matching) | 0.0 | 0.0 | 46.1 | 46.1 | 7.0 | 1.4 | 18.8 | 17.2 | 17.2 | 1.4 | 9.4 | 7.0 | **91.4** | 1.4 | 3.1 | 0.0 | 30.3 | 1.6 | 19.3 | 17.6 |
| ImageNet SwinT-B-IN1K | BIM (Logit Matching) | 0.8 | 1.6 | 46.1 | 45.3 | 7.0 | 2.0 | 7.0 | 7.0 | **55.5** | 2.0 | 0.8 | 0.8 | **90.6** | 0.2 | 0.0 | 0.0 | 41.2 | 2.0 | 13.5 | 13.1 |

Table 4: Results of Logit Matching as adaptive attack against our method.

**Interpolation with Binary Search.** (from Section 5.13 "Turning a Weakness into a Strength" of [13]) This methods find interpolated adversarial examples that are close to the decision boundary

with binary search. We conduct experiment with all testing data from CIFAR-10, and 128 random testing samples from ImageNet (due to limited time frame of rebuttal). The experimental results can be found in the following table. Compared to the baseline results, the results show that our method is still effective against the adversarial examples close to the decision boundary.

| Dataset Model | Attack | $\epsilon = 2/255$ | | | | $\epsilon = 4/255$ | | | | $\epsilon = 8/255$ | | | | $\epsilon = 16/255$ | | | | $\epsilon = ?$ | | | |
|---|---|---|---|---|---|---|---|---|---|---|---|---|---|---|---|---|---|---|---|---|---|
| | | DR | FPR | Acc | Acc* | DR | FPR | Acc | Acc* | DR | FPR | Acc | Acc* | DR | FPR | Acc | Acc* | DR | FPR | Acc | Acc* |
| CIFAR-10 ResNet-18 | BIM (Interpolation) | 0.0 | 0.0 | 65.7 | 65.7 | 0.0 | 0.0 | 44.6 | 44.6 | 28.0 | 1.5 | 21.9 | 28.0 | **74.4** | 1.1 | 6.0 | 56.4 | 28.0 | 1.5 | 34.6 | 48.8 |
| ImageNet ResNet-152 | BIM (Interpolation) | 0.0 | 0.0 | 18.8 | 18.8 | 4.7 | 1.4 | 6.2 | 5.5 | 25.0 | 1.4 | 0.8 | 0.8 | **90.6** | 1.4 | 0.0 | 0.8 | 31.4 | 1.6 | 6.4 | 6.2 |
| ImageNet SwinT-B-IN1K | BIM (Interpolation) | 1.6 | 1.6 | 44.5 | 45.3 | 3.9 | 2.0 | 37.5 | 35.9 | **66.4** | 2.0 | 14.1 | 64.8 | **97.7** | 0.2 | 0.0 | 97.7 | 42.8 | 2.0 | 24.0 | 61.3 |

Table 5: Results of Interpolation with Binary Search as adaptive attack against our method.

**Adaptive Attack discussed in Section 2.** To avoid triggering SAE, the goal of the PGD attack can include an additional term to minimize $\|\boldsymbol{S}_*(\boldsymbol{x} + \boldsymbol{r})\|_F$. Namely, the corresponding adaptive attack is:

$$\arg\max_{\boldsymbol{r}} L_{\text{CE}}(\boldsymbol{x} + \boldsymbol{r}, c(\boldsymbol{x})) - \|\boldsymbol{S}_*(\boldsymbol{x} + \boldsymbol{r})\|_F$$

$$= \arg\max_{\boldsymbol{r}} L_{\text{CE}}(\boldsymbol{x} + \boldsymbol{r}, c(\boldsymbol{x})) - [\sum_i \sum_j |s_{n^*}^{(i,j)}|^2]^{1/2}$$

$$= \arg\max_{\boldsymbol{r}} L_{\text{CE}}(\boldsymbol{x} + \boldsymbol{r}, c(\boldsymbol{x})) - [\sum_{i=1}^{T+1} \sum_{j=1}^{T+1} \cos[\nabla f_{n^*}(\boldsymbol{x} + \boldsymbol{r} + \boldsymbol{\delta}_i), \nabla f_{n^*}(\boldsymbol{x} + \boldsymbol{r} + \boldsymbol{\delta}_j)]^2]^{1/2}$$

To solve this adaptive attack problem, the straightforward solution is to conduct $Z$-step PGD updates with the modified loss function. Each step includes but is not limited to these computations: (1) $T + 1$ Jacobian matrices to calculate $n^*$ and $\nabla f_{n^*}(\cdot)$; (2) $T + 1$ Hessian matrices to calculate $\nabla^2 f_{n^*}(\cdot)$. Let $\psi_J$ and $\psi_H$ be the time consumption for Jacobian and Hessian matrices respectively. Then the time consumption of the $Z$ steps of optimization in total is greater than $Z(T + 1)(\psi_J + \psi_H)$.

For reference, for Nvidia Titan Xp GPU and CIFAR-10/ResNet-18, the $\psi_J = 0.187 \pm 0.012$ seconds, and $\psi_H = 20.959 \pm 0.679$ seconds (Python code for this benchmark can be found in Appendix). If we use $Z = 100$ steps of PGD attack, and $T = 6$ for calculating ARC, each adversarial example of a CIFAR-10 image takes more than $Z(T + 1)(\psi_J + \psi_H) \approx 14802$ seconds (i.e., 4.1 hours).

Note, we acknowledge that other alternative adaptive attack designs are possible, but as long as the alternative design involves optimizing any loss term calculated from gradients, second-order gradients (Hessian) will be required to finish the optimization process, which again makes the alternative attack computationally prohibitive.

## A.5 More on Related Works

We discuss the related works in more details, as an extension to Section 6.

**Similar Defenses.**

- "The Odds are Odd" [30] is an attack detection method based on feature statistical test. This method is categorized in Section 6 as feature statistics-based methods. In particular, it detects adversarial examples based on the difference between the logits of clean image and image with random noise. This method assumes that a random noise may break the adversarial perturbation and hence lead to notable changes in the logits, and is is capable of correcting test time predictions. Meanwhile, it can be broken by adaptive attack to match the logits with an image from another example [13]. Similarly, our method can be seen as a statistical test for gradient consistency as reflected by ARC feature. Our method is motivated by the assumption that neural networks will manifest "local linearity" with respect to adversarial examples, which will not happen for benign examples. Meanwhile the SAE is consistent across different architectures, and the corresponding 2-D ARCv feature shows very simple cluster structure for both benign and adversarial examples. The adaptive attack against [30] can merely slightly reduce the effectiveness of our attack, as shown in the additional adaptive attack experiments in this Appendix.
- "Turning a Weakness into a Strength" [56] is an attack detection method which is conceptually similar to [30]. This method involves two criterion for detection: (1) low density

of adversarial perturbations – random perturbations applied to natural images should not lead to changes in the predicted label. The input will be rejected if the change in predicted probability vector is significant after adding a Gaussian noise. (2) close proximity to the decision boundary – this leads to a method that rejects an input if it requires too many steps to successfully perturb with an iterative attack algorithm. Hence, this method can be seen as an detector with two-dimensional manually crafted feature. This method can be broken by an adaptive attack [13] that searches for an interpolation between the benign and adversarial example. Similarly, our method leverages BIM, an iterative attack to calculate the ARC feature. However, differently, our method use the iterative attack to explore the local area around the input, in order to calculate the extent of "local linearity" around the point as the ARC feature, while [56] leverages an iterative attack to count the number of required steps. The ARC feature shows clear difference between benign and adversarial examples, and hence does not need to combine with other manually crafted feature. [56] points out that solely using one criterion is insufficient, because the criterion (1) may be easily bypassed. The adaptive attack against [56] can merely slightly reduce the effectiveness of our attack, as shown in the additional adaptive attack experiments in this Appendix.

**Local Linearity.** Local linearity is an important characteristics for the community to understand the adversarial attack as well as design defense methods.

- FGSM [6] is designed based on the intuition that neural networks are vulnerable because their "local linear" property has been triggered by the attack. This is the first work that propose the concept of "local linearity" about adversarial attack. Many follow-up works about "local linearity" are adversarial training methods.

- In LLS [27] (adversarial training), a regularizer is proposed that encourages the loss to behave linearly in the vicinity of the training data, thereby penalizing gradient obfuscation while encouraging robustness. This is relevant to our interpretation on adversarial training in Section 5.

- In GradAlign [57] (adversarial training), it is noted that the network being highly non-linear locally is the main reason why FGSM training fails.

- Sparsifying front end [58] points out that a "locally linear" model can be used to develop a theoretical foundation for crafting attacks and defenses.

- In [59], it is proved that the Fast Gradient Method attack and a Randomized Smoothing defense form a Nash Equilibrium, under a locally linear decision boundary model for the underlying binary classifier.

- [60] shows that local linearity arises naturally at initialization.

### A.6 Python Code for Evaluating Time Consumption of Jacobian / Hessian Calculation

The python code for measuring the time consumption for Jacobian and Hessian matrices calculation is shown below. The code is based on CIFAR-10 settings with $M = 3 \times 32 \times 32$ and $N = 10$, and the neural network used is ResNet-18. For reference, the result on Nvidia Titan Xp GPU is $0.187 \pm 0.012$ seconds for Jacobian, and $20.959 \pm 0.679$ seconds for Hessian.

Note, for the ImageNet/ResNet-152 case, the Jacobian and Hessian calculation cost is much higher.

```python
import time, torch as th, torchvision as V, numpy as np
device = 'cuda'
resnet18 = V.models.resnet18(False).to(device) # standard resnet18
resnet18.eval()
resnet18.fc = th.nn.Linear(512, 10).to(device) # fit for 10 classes
X = th.rand(1, 3, 32, 32).to(device) # random input
# compute a jacobian
time_start = time.time()
J = th.autograd.functional.jacobian(resnet18, X)
time_end = time.time()
```

```
print('A Jacobian takes:', time_end - time_start, 'seconds')
# compute a hessian
time_start = time.time()
H = th.autograd.functional.hessian(lambda x: resnet18(x)[0, 0], X)
time_end = time.time()
print('A Hessian takes:', time_end - time_start, 'seconds')
```

