# OpenReview forum: "On Trace of PGD-Like Adversarial Attacks"
_NeurIPS.cc/2022/Conference — NeurIPS 2022 Submitted_

### Official Review · Reviewer_YXrD · 2022-06-14

**Rating:** 4
**Confidence:** 4
**Soundness:** 3 good
**Presentation:** 3 good
**Contribution:** 2 fair

**Summary:**

The authors propose a new detection method to detect PGD-like attacks. The detection method is based on the analysis of local linearity between benign samples and adversarial samples. The authors also introduce ARC and SAE to measure the local linearity. The experimental results indicate that this method can successfully detect PGD-like attacks on various datasets and model architectures.

**Questions:**

1. The authors claim that this method is only sensitive to PGD-like attacks. Is it because that only PGD-like attacks can trigger the local linearity? If a PGD-like attack does not trigger it, can the detection still sensitive to this attack?
2. In Figure 5, what is the reason that NSS is a better detection method on ResNet-152?

**Limitations:**

The authors have addressed the limitations. I appreciate it.

**Strengths And Weaknesses:**

I think using gradients to detect AEs is interesting, and there are few works focusing on this point. However, there are some weaknesses.

1. The design of adaptive attack is not convincing. To minimize the Fro-norm of S_*(x+r), the attacker only needs to adopt an additional loss term to minimize the difference between the gradients of AEs and clean data.
2. There are five requirements for SAE to be consistently triggered. It seems that breaking requirements (III), (IV) and (V) are very easy. The threat model is too strict for me. The victim needs to have perfect knowledge of the attacker.
3. As a detection method, its premier requirement is fast and cannot influence the performance (e.g., the predicting speed and accuracy) of the classifier. This detection is too slow to use in practice.
4. The DR in Table 2 is very low, even for the baseline (t1).
5. For ImageNet, the perturbation size is usually very small, i.e., 4./255 or less. So, the results in Table 3 indicate that this detection method cannot efficiently detect AEs for ImageNet.
6. The baseline method is only NSS, which is not enough. Although the proposed method is data-undemanding, I think a victim can freely gather as much as clean data to train a detector. It should not be a problem to compare your method with other previous works.

---

> ### Author Response · Authors · 2022-07-31
> **Detailed justifications.**
>
> **1. Design of adaptive attack not convincing**
>
> We acknowledge that other alternative adaptive attack designs are possible, but as long as a loss term involves gradients, second-order gradients (Hessian) will be inevitable for optimization, which makes it computationally prohibitive again. We also discuss other adaptive attacks mentioned by other reviewers in Appendix A.
>
> **2. Threat model too strict**
>
> Our method inherits limitations from the strong assumptions, which are justified in Section 4.2. Besides, for forensics, our method can identify attack type (whether or not PGD-like) and infer attack details, while existing detectors cannot distinguish results created by different attacks. This is less explored in the literature.
>
> **3. This detection is too slow**
>
> Our method is very slow due to Jacobian matrix calculation (Section 5 “Limitations” in initial submission). The proposed method is slow for real-time defense, but is still suitable for forensics. We summarize the pros and cons in Appendix.
>
> **4. The DR in Table 2 is very low, even for the baseline (t1)**
>
> The effectiveness of our method relies on the five conditions disucssed in Section 2. In Section 4.2, we justify these conditions by breaking them respectively as shown in Table 2. Hence, Table 2 includes the baseline (t1) and the cases (all except for t1) under which our method will not be very effective due to clearly broken assumption. Namely, high DR in Table 2 is not expected and will be paradox.
>
> The expeirments are carried out with the most difficult “epsilon=?” setting (more difficult than the common settings in related works), which involves many small perturbations that are hard to detect. Hence the baseline (t1) performance is not high.
>
> **5. The results in Table 3 indicate that this detection method cannot efficiently detect AEs for ImageNet**
>
> Smaller perturbations are harder to detect. The unique trace of PGD-like attacks is stronger when the attack is stronger, and hence the detector performance will be better with a stronger trace (98% DR for ImageNet/SwinT with epsilon=16/255). This characteristics is consistent across different settings, and none of the existing methods shows the same characteristics. For instance, the DR of NSS even drops by a large margin when epsilon is increased from 8/255 to 16/255, which is an undesired characteristics showing inconsistency of NSS.
>
> **6. The victim should be able to freely gather as much as clean data to train a detector**
>
> Data-demanding methods are only applicable for models using publically available datasets, or is only applicable by the ones who trained the neural network. This seriously limits the use cases of these methods, while a data-undemanding method can be used in a wider range of defense or forensics scenarios. Collecting a large amount of data may be difficult for some potential adopters. For instance, the raw training data of a federated learning face recognition model is inaccessible (sending raw image data is forbidden in Federated Learning due to privacy violation), but collecting a few reference samples from volunteers is still possible. None of data-demanding methods is applicable in this case.
>
> Our proposed method focuses on the network response characteristics. It does not benefit much from a large amount of training data. As discussed on L297-301, the performance gain of our method starts to plateau from roughly 200 training samples.
>
> **7. This method is only sensitive to PGD-like attacks. (followed by two questions)**
>
> Our ARC feature is calculated using BIM, in order to “continue” a previous PGD-like attack (Section 2). Hence, SAE relies on the five conditions in Section 2, and non-PGD-like attack will easily break the conditions. Our empirical observations (all Figures and Tables) show that only PGD-like attacks (will easily satisfy the conditions) will effectively trigger SAE (local linearity).
>
> Statistically, PGD-like attacks will trigger local linearity based on our demonstrations (Figure 2), but there will be hard samples that fall into the cluster of benign examples. Our method is not sensitive against these samples as demonstrated, which leads to imperfect Detection Rate.
>
> **8. In Figure 5, what is the reason that NSS is a better detection method on ResNet-152?**
>
> ARC is based on model gradient, and different architecture have different characteristics. Compared to SwinT, ResNet-152 is harder to be turned “linear” (Figure2; Figure3; L235). We speculate this is the reason why the trace for ResNet-152 is weaker than that of SwinT.
>
> Figure 5 is based on the difficult “epsilon=?” setting involving both small and large perturbations. According to Table 3, a larger perturbation results in stronger trace, and hence more distinct SAE and higher DR for our method. NSS extracts low level hand-crafted features, and is better than our method for small perturbations. But NSS performance will plateau or even decrease with larger perturbations, which is inconsistent.

---

> > ### Comment · Reviewer_YXrD · 2022-08-05
> > **Comments after reading rebuttals**
> >
> > Thanks for the answers. There are some comments from me after reading the rebuttals.
> >
> > Firstly, for the adaptive attacks and the authors' reply to Question 4. ''The DR in Table 2 is very low, even for the baseline (t1)'', the adversary can break the conditions to decrease the detection success rate. Clearly, these attacks can be seen as adaptive attacks.
> >
> > Secondly, for the detection speed, the authors claim that their method can be used in forensics. However, in the experiments, there is no evidence that their method is better than existed forensics methods.
> >
> > Finally, for the generalizability, based on the authors' reply, the method is not general. The model structures, data distributions, and attack algorithms will influence the detection performance significantly.

---

> > > ### Author Response · Authors · 2022-08-05
> > > **Further discussion**
> > >
> > > We appreciate further comments from the reviewer.
> > >
> > > We acknowledge that there are many choices of adaptive attacks against the proposed method, and it suffers from slow computation largely due the the computation of Jacobian matrices. Meanwhile, the proposed method is specific to PGD-like attacks (as reflected by title). However, it shows a consistent pattern that the unique trace (SAE as reflected by ARC features) becomes stronger when the attack is stronger across different architectures, although the concrete performance differs across architecture.
> > >
> > > We would like to further emphasize two differences between the proposed method and related works:
> > >
> > > **(1)** Our problem setting requires the least amount of information from the potential user (only a pre-trained network as well as a few training samples), which makes the method applicable in a wider range of scenarios, including scenarios where most existing methods will be infeasible (e.g., Federated Learning as discussed). This problem setting is extremely challenging, and is less explored in the literature.
> > >
> > > **(2)** While existing detectors can detect multiple types of attacks, they can only infer whether the input is adversarial. While our proposed method is specific to PGD-like attacks, it can infer more information about the attack including loss, label, and Lp bound, etc., based on the five conditions.

---

### Official Review · Reviewer_vEc7 · 2022-07-07

**Rating:** 6
**Confidence:** 5
**Soundness:** 3 good
**Presentation:** 3 good
**Contribution:** 2 fair

**Summary:**

This paper proposes a detection defense against adversarial examples generated by untargeted PGD-like attacks. This defense relies on the assumption that such adversarial examples trigger distinguishable patterns of linearity along a local trace defined by the BIM attack, which maximizes the cross-entropy loss that regards the least-likely label as the ground truth. The linearity pattern is characterized by the ARC feature, a vector reduced from the jacobian matrices along the BIM trace. Extensive qualitative and quantitive results on CIFAR10 and ImageNet (ResNet and SwinT) demonstrate the distinguishability of the ARC feature over benign inputs and adversarial examples generated by PGD-like attacks. This characterization is then used to detect PGD-like attacks and infer their parameters. The overall defense is effective against non-adaptive (unaware of the defense) PGD-like attacks, and outperforms another detection defense that does not require too much data to collect statistics. Finally, some interesting observations are discussed, such as linearity enforced by adversarial training.

**Questions:**

My current score is based on the strengths of the proposed characterization method and the expected fix of minor weaknesses noted above. I am willing to raise my score if the following major concerns are adequately clarified or justified.
* [Quality-Weakness-1] What if the least-likely label is not the ground truth label?
* [Quality-Weakness-2] Discuss if the defense is indeed lightweight compared with previous defenses requiring auxiliary models.
* [Quality-Weakness-3.1] Discuss the computational cost of the proposed adaptive attack on CIFAR-10.
* [Quality-Weakness-3.2] Discuss the two adaptive attacks mentioned above.

I am open to decrease the significance of the last question if the meta-reviewer deemed it as beyond scope or outweighed by the strengths of the proposed characterization method in a non-adversarial setting.

**Limitations:**

Since this paper claims to propose a defense, its main limitation is the lack of a sufficient discussion of adaptive attacks. Although the authors have provided some discussion at L123-134, I find it hard to be convinced that a more thorough evaluation is not necessary. It is strongly recommended to at least evaluate published adaptive attacks (that break previous similar defenses) on the proposed defense.

**Strengths And Weaknesses:**

Before heading to the detailed comments, I would like to note *neutrally* that the proposed defense is similar in spirit to some previous detection defenses that rely on the adversarial example's higher robustness against benign noise [1] and adversarial perturbation [2], which were later broken by [3]. *This similarity leads to both pros and cons, detailed below.*

[1] The Odds are Odd: A Statistical Test for Detecting Adversarial Examples. Roth et al. ICML 2019.

[2] A New Defense Against Adversarial Images: Turning a Weakness into a Strength. Hu et al. NeurIPS 2019.

[3] On Adaptive Attacks to Adversarial Example Defenses. Tramèr et al. NeurIPS 2020.

### Originality

**Strengths (major)**
* The proposed characterization of adversarial examples is novel and only requires a tiny set of training data, which seems to be less discussed in the literature.
* The proposed ARC feature is novel and explicitly characterizes a well-defined property (local linearity) of adversarial examples. I believe this is more sophisticated than previously explored properties such as robustness to benign noise [1] and adversarial perturbation [2].
* The idea of using BIM attacks to characterize the trace of untargeted PGD-like attacks is insightful. The proposed defense can detect perturbed-but-unsuccessful inputs, which seemed not to be covered by previous defenses [1, 2].

**Weaknesses (minor)**
* **Discuss similar defenses.** It is suggested to discuss [1, 2] and potentially other similar defenses with more details on the underlying properties. For example, how the property explored in this work is different and superior to those explored by previous defenses.
* **Discuss local linearity.** Since this work relies on the local linearity of adversarial examples, it is suggested to include more discussion about newer work in this direction to solidify the underlying assumption.

### Quality

**Strengths (major)**
* The proposed characterization of adversarial examples is technically sound and well supported by experimental results on CIFAR-10 (ResNet) and ImageNet (ResNet and SwinT). It can effectively distinguish between benign inputs and adversarial examples generated by PGD-like attacks with different parameters.
* The ablation study demonstrates the defense's non-sensitivity to non-PGD-like attacks, but the pros and cons are adequately justified.

**Weaknesses (major)**

* **Some assumptions are too strong.** This paper made some strong assumptions that may not always hold. For example, at L81 the authors assume that the input's ground-truth label is simply the least-likely one. While this is likely to hold for vanilla attacks, a slightly smarter attacker could simply make the ground-truth label anywhere between the most-likely and least-likely label to break this assumption. It is unclear how the original characterization performs in this case. For example, is it possible that the linearity goes down at the first few steps (given an incorrect guess of ground truth) before heading up in those cases?

* **The proposed defense may not be lightweight**. Since the proposed defense requires several steps of BIM attacks at the inference stage to obtain the input's ARC feature, the defended model's inference of each input would now include overheads (both efficiency and memory) of several backward passes. While other defenses require auxiliary models (thus argued as not lightweight in this paper), those defenses do not involve backward passes and thus may be more efficient than the proposed defense. This weakness is partially supported by the authors' inability to evaluate all ImageNet data. It is suggested to include some empirical results to resolve or clarify this weakness.

* **Insufficient discussion of adversarial robustness.** Since this paper claims to propose a defense against adversarial examples, I find it hard to be convinced that the proposed defense would come with much robustness against adaptive attacks. I understand that the authors have discussed the hardness of adaptive attacks to some extent and left them for future study (L123-134), but the current discussion is rather limited, even from the perspective of *published* adaptive attacks on similar defenses. *I am outlining my comments below but will be open on this point if the meta-reviewer deemed the following discussion as beyond the scope or outweighed by the strengths of the proposed characterization approach (in a non-adversarial setting).*
  * At L123-134, the authors discussed an adaptive attack that is computationally prohibitive on ImageNet. However, it is unclear if the same claim holds for the smaller CIFAR-10, which is also a dataset evaluated in this paper. **Please discuss the computational cost of this adaptive attack on CIFAR-10 and if it is still prohibitive.**
  * If I understand correctly, the adaptive attack must reach a point whose BIM-trace is equally non-linear (or has the same linearity pattern) as that of a benign input. **However, the current defense cannot prevent the existence of such points, so it is still possible to find them.** This is similar to previously broken defenses that expect adversarial inputs to have unique patterns in terms of robustness to benign noise [1] and adversarial perturbation [2]. Therefore, I believe discussing their *existing* adaptive attacks from [3] would significantly strengthen this paper. In particular, it is suggested to discuss the following two *published* adaptive attacks.
    * Logit matching (Section 5.2 of [3]). If the PGD attacker now aims to reach a point whose logit is sufficiently close to an unperturbed image from a different class, can the proposed characterization still distinguish between benign inputs and such adversarial examples?
    * Interpolation between the adversarial and benign examples (Section 5.13 of [3]). If the attacker moves the adversarial example generated by PGD towards the original benign input by interpolation, can the proposed characterization still distinguish between the original benign input and the adversarial example moved towards it?


### Clarity

**Strengths (major)**
* The overall presentation is good. I appreciate the clear demonstration in Figure 1.
* The summarized conditions in Section 4.2 is good.

**Weaknesses (minor)**
* The setting discussed at L24-26 comes out without any context, making it hard to understand why it is important and hard to achieve. It is suggested to motivate this setting with some related work and emphasize it more throughout the paper.
* Similarly, at L59 I can see that the setting is extremely limited, but what are the strong "cues" and why are they hard to solve?

### Significance

I appreciate the great effort in characterizing the trace of PGD-like attacks and the extensive experiments; this paper might inspire some work in that direction. However, my biggest concern about this work is its robustness as a defense against adversarial examples, given my experience of the commonly acknowledged importance of evaluating adaptive attacks in the adversarial example defense literature. That being said, I am open to discussion on this point.

---

> ### Author Response · Authors · 2022-07-31
> **Detailed Justifications (thanks for the very detailed review)**
>
> **Quality-1. What if the least-likely label is not the ground
> truth label? Is it possible that the linearity goes down ...?**
>
> This exactly corresponds to the experiments for our condition (V)
> introduced in Section 2. This condition is justified in Section 4.2.(V):
> PGD-like attacks triggers local linearity for a considerable portion
> of network output dimensions, because there will be SAE even with
> a randomly guessed label (f10 in Figure 4). The best guess leads to
> the strongest SAE (f1 in Fig 4), while the worst guess leads to the
> least significant SAE (f9 in Fig 4).
>
> Besides, the linearity will not "go down", as shown in (f10) of
> Figure 4 using a randomly guessed label -- the values in matrix are
> increasing. Namely, the linearity "goes up'' with any guess (most-likely,
> least-likely, random) according to visualizations mentioned in Section
> 4.2.(V).
>
> **Quality-2. Discuss if the defense is indeed lightweight compared
> with previous defenses requiring auxiliary models.**
>
> Sorry for the ambiguity. The attribute "light-weighted'' is justified
> in Section 1 "Contributions'': "light-weighted (requires no auxiliary
> deep model)''. Namely, it is light-weighted in terms of algorithm
> components. The proposed method is slow due to Jacobian
> calculation, which is justified in Section 5 "Limitations'' in
> initial submission. This is also included in the "Pros\&Cons''
> list in the Appendix.
>
> **Quality-3.1. Discuss the computational cost of the proposed
> adaptive attack on CIFAR-10.**
>
> The additional loss term `||S_*(x+r)||_F` can be expanded with
> Eq.(3) as shown in Appendix A.4. To solve this adaptive attack problem,
> a straightforward solution is to conduct $Z$-step PGD updates with
> the additional loss term. Thus, each step includes but is not limited
> to these computations: (1) $T+1$ Jacobian matrices to calculate $n^{\*}$
> and $\nabla f_{n^{\*}}(\cdot)$; (2) $T+1$ Hessian matrices to calculate
> $\nabla^{2}f_{n^{\*}}(\cdot)$. Let $\psi_{J}$ and $\psi_{H}$ be
> the time consumption for Jacobian and Hessian matrices respectively.
> Then the time consumption of the $Z$ steps of optimization in total
> is greater than $Z(T+1)(\psi_{J}+\psi_{H})$.
>
> For reference, for Nvidia Titan Xp GPU and CIFAR-10/ResNet-18, the
> $\psi_{J}=0.187\pm0.012$ seconds, and $\psi_{H}=20.959\pm0.679$
> seconds (Python code for this benchmark can be found in Appendix).
> If we use $Z=100$ steps of PGD attack, and $T=6$ for calculating
> ARC, each adaptive adversarial example of a CIFAR-10 image takes more
> than $Z(T+1)(\psi_{J}+\psi_{H})\approx14802$ seconds (i.e., $4.1$
> hours).
>
> **Quality-3.2. Adaptive attacks: Logit matching and Interpolation
> with binary search.**
>
> For
> CIFAR-10, we use all testing data. For ImageNet, we only use 128 samples
> due to limited time frame. The detailed results and discussions are
> added to Appendix A.4.
>
> For __Logit Matching__, the (DR, FPR) at epsilon=\{2,4,8,16,?\}/255 are
> respectively:
>
> (0.0, 0.0), (0.0, 0.0), (23.8, 1.5), (48.0, 1.1), (22.8, 1.5) for
> ResNet-18;
>
> (0.0, 0.0), (7.0, 1.4), (17.2, 1.4), (91.4, 1.4), (30.3, 1.6) for
> ResNet-152;
>
> (0.8, 1.6), (7.0, 2.0), (55.5, 2.0), (90.6, 0.2), (41.2, 2.0) for
> SwinT.
>
> For __Interpolation__ method, the corresponding (DR, FPR) are:
>
> (0.0, 0.0), (0.0, 0.0), (28.0, 1.5), (74.4, 1.1), (28.0, 1.5) for
> ResNet-18;
>
> (0.0, 0.0), (4.7, 1.4), (25.0, 1.4), (90.6, 1.4), (31.4, 1.6) for
> ResNet-152;
>
> (1.6, 1.6), (3.9, 2.0), (66.4, 2.0), (97.7, 0.2), (42.8, 2.0) for
> SwinT.
>
> The SAE is expectedly weaker with the two attacks compared to the
> baselines in Table 1, but our method still remains effective.
>
> **Clarity-1. Motivation of the setting discussed at L24-26 comes
> out without any context, making it hard to understand why it is important
> and hard to achieve.**
>
> An extremely limited problem setting makes the proposed method valid
> and feasible in a wider range of defense and forensics scenarios.
> This is discussed in detail in Appendix A.3. The most straightforward
> example is face recognition model with Federated Learning. It is impossible
> to access the raw training data from user device (violating privacy
> of a large number of users), but it is still possible to collect 50
> samples from several volunteers. In this case, data-demanding methods
> will be infeasible, while our method is still valid.
>
> **Clarity-2. Similarly, at L59 I can see that the setting is
> extremely limited, but what are the strong "cues"
> and why are they hard to solve?**
>
> The "strong cues'' is a transitional sentence in order to smoothly
> introduce our method. We will change the sentence into ``requiring
> us to identify strong traces left by the adversarial attacks'' to
> avoid confusion. This problem is hard because it requires the least
> amount of information from the user to detect adversarial examples
> compared to related works.
>
> **Originality-\{1,2\}. Discuss similar defenses. Discuss local
> linearity.**
>
> We added the corresponding discussions in the Appendix. The discussion
> is omitted here due to character limit.

---

> > ### Comment · Reviewer_vEc7 · 2022-08-05
> > **Response to Authors**
> >
> > Thank you for your response. It is good to see that
> > 1. The adaptive attack’s cost on CIFAR10 is still high
> > 2. The defense has reasonable robustness to previous adaptive attacks
> > 3. Discussion of similar defenses
> >
> > Given this, I am raising my score from 5 to 6.
> >
> > ---
> >
> > I could not give 7 due my only remaining concern, that **the applicable scenario of this method is somewhat limited.** This concern arose as I went through other reviewers' comments on the strong threat model.
> >
> > Below is my best disentanglement of this issue.
> >
> > **1. This work as a non-defensive method to characterize PGD-like attacks.**
> >
> > As indicated in my original review, I like the proposed method of characterizing adversarial examples created by PGD-like attacks. I believe this method should inspire some work in this direction, i.e., characterizing adversarial examples and inferring the underlying attack settings.
> >
> > **Pros:** From this perspective, it is OK to have strong assumptions (i.e., the SAE conditions), as they benefit inferring the attack details. That means, being sensitive to these conditions is a strength.
> >
> > **Cons:** The problem is, I am not sure if there are many practical scenarios where someone wants to characterize white-box attacks (i.e., the PGD-like attacks considered in this paper). It might be good to provide some justifications about why this method is needed in practice. (sorry for brining up this new information not indicated in the original review)
> >
> > **2. This work as a defense (with robustness claims) to detect PGD-like attacks.**
> >
> > **Pros:** The defense is robust to a reasonable set of adaptive attacks, assuming that the strong threat model is reasonable.
> >
> > **Cons:** From this perspective, however, I am afraid it is NOT OK to have strong assumptions, as attacks can easily break some of them to evade the defense. That means, being sensitive to these conditions is a weakness.
> >
> > **3. Summary**
> >
> > I believe the above disentanglement illustrates my root concern for this paper — it is good as a non-defensive characterization but is bad as a defense.
> >
> > From responses to other reviewers, I can see that the authors have been trying to justify the validity of their threat model from the first perspective. However, they should notice that the claims as a defense belong to the second perspective, where the threat model is surely too strong. I would suggest the authors carefully think about what is the best way to present this method, as the current claims are slightly convolved.

---

> > > ### Author Response · Authors · 2022-08-05
> > > **Further discussion**
> > >
> > > We fully agree with the reviewer's summary. Specifically, the two mentioned perspectives (i.e., “non-defensive characterization” and “defense”) are exactly Section 2: “Adversarial Response Characteristics & Sequel Attack Effect” and Section 3: “Attack Detection and Inferring Attack Details”. They are organized logically, and we also realize that some minor modifications are required to better balance the emphasis on the two Sections based on the discussions with reviewers.
> > >
> > > As reflected by the title “On Trace of PGD-Like Adversarial Attacks”, the core part of this manuscript is Section 2, namely the non-defensive characterization of PGD-like attacks. The qualitative demonstrations in Figure 2 is intriguing and demonstrate the effectiveness of the proposed characterization method. But there is not yet way to quantitatively support it. Based on this, a direct application of such characterization (such as attack detection) can serve as a quantitative support for the proposed characterization. And hence attack detection is organized into an isolated section.
> > >
> > > In order to better present the proposed method, we will make the following minor revisions to better highlight the characterization and make the logic of the manuscript less confusing:
> > >
> > > **(1)** From Abstract to Section 1: we will stick to the title and clarify the non-defensive characterization is the core of the manuscript. And we will explicitly justify that attack detection is a direct application of the proposed characterization method, which serves as a quantitative evaluation of the characterization method.
> > >
> > > **(2)** Section 3: we will explicitly move the “Limitations” paragraph from Section 5 to the end of Section 3, and meanwhile include the limitations discussed with reviewers in the revised paragraph. In this way there will be a smaller risk of confusing future readers – the proposed method is a non-defensive characterization method, but will face the discussed limitations when really used for defense.
> > >
> > > **(3)** We will reduce the claims on defense throughout the manuscript, and will explicitly clarify the motivation of providing Section 3 (quantitative support for the characterization method) at the beginning of Section 3.
> > >
> > > In the end, thanks again for the very constructive comments.

---

> > > ### Author Response · Authors · 2022-08-05
> > > **Why white-box attacks (PGD-like attacks)?**
> > >
> > > PGD-like attacks are rather common in the literature on attack/defense. Meanwhile, PGD is also a part of standard emprirical evaluation tool for adversarial robustness. Sharing the characterization method with the community may inspire future works involving:
> > >
> > > **(1)** similar attack detection problems. For example, how to detect black-box attacks (e.g., score-based like NES/SPSA, or transfer-based like TI-FGSM/DI-FGSM) under the same extremely challenging problem setting? Characterizing black-box attacks is more practical. But since these attacks involve more uncertain factors (these attacks behave much more randomly than PGD-like attacks), characterizing these attacks could be even more challenging under our problem setting, compared to the well-conditioned PGD-like attacks.
> > >
> > > **(2)** stronger attacks. The characterization of the PGD-like attacks show their difference compared to other non-PGD-like attacks such as C&W. Future works may be inspired based on further analysis on the differences in characteristics.
> > >
> > > **(3)** stronger adversarial training. PGD-like attacks is commonly used for robustness evaluation. Our PGD characterization and the new interpretation on defense (Section 5 “Combination with Adversarial Training”) may inspire future defense works.

---

### Official Review · Reviewer_nyro · 2022-07-11

**Rating:** 6
**Confidence:** 4
**Soundness:** 4 excellent
**Presentation:** 4 excellent
**Contribution:** 3 good

**Summary:**

This paper proposes a new and interesting method to detect PGD-like adversarial perturbations. The insight is to leverage the fact that PGD-like adversarial attacks will trigger local linearity of the input sample. To quantitatively measure the local linearity, the authors propose to use the Sequel Attack Effect (SAE) to continue the attack with BIM and then measure linearity with Adversarial Response Characteristics (ARC). SVM-based models are then trained and used for the detection of informed/uninformed attack detections.

**Questions:**

- I understand the main contribution here is for detection with limited amount of training data. I wonder how well the performance of the proposed method would be given sufficient training data compared to other SOTA detection methods?
- A minor question: why use T=48 in line 100? It would be better to give a brief explanation for special numbers.

**Limitations:**

Limitations and societal impact are comprehensively discussed in the paper.

**Strengths And Weaknesses:**

Strength
- The proposed method is very interesting and novel, providing great contributions to understanding and detecting PGD-like adversarial attacks and defense.
- The investigated topic is important and useful.
- Extensive empirical results on CIFAR-10 and ImageNet well illustrate the effectiveness of the proposed method and support the claims.
- The paper is well-written and easy to follow. Provided figures clearly show the insights of the method.

Weakness
- The method is only limited to PGD-like adversarial perturbations, unable to other popular attacks like C&W as discussed in the paper.
- The importance of limited training data for adversarial detection should be better motivated in introduction.

---

> ### Author Response · Authors · 2022-07-30
> **Data-undemanding methods can be used in a wider range of scenarios where employing data-demanding methods is impossible, such as third-party attack detection / forensics, as well as federated learing.**
>
> **1. The proposed method is limited to PGD-like attacks as discussed in the paper.**
>
> The proposed method relies on the five assumptions as discussed in “Uniqueness of SAE to PGD-like Attack” of Section 2, which are further justified in Section 4.2. These assumptions collectively make the proposed method specific to PGD-like attacks. Being insensitive to non-PGD attacks makes the proposed method unsuitable for general attack detection scenario, but it is still useful in defense scenario with knowledge about the attacker, or forensics scenario to example whether an adversairla example is created by PGD-like attack by identifying its unique trace.
>
> We summarized a list of pros and cons of the proposed method in the Appendix of the revised manuscript.
>
> **2. Motivation for extremely limited training data.**
>
> Data-demanding methods is only applicable for models using publically available datasets, or is only applicable by the first-party who trained the neural network. This limits the use cases of these methods. In contrast, we do not assume collecting a large amount of data is easy for potential adopters of the proposed method. Due to the low demand on data, the proposed method enables a wider range of defense or forensics scenarios, especially when there is no access to the whole training dataset. For instance, the __"Third-party Attack Detection or Forensics"__ and __"Attack Detection for Federated Learning"__ scenarios.
>
> __Third-party Attack Detection (identify whether the model is attacked) or Forensics (identify attack type and infer the attack detail).__ Being data-undemanding means the proposed method can be applied to any pre-trained neural network randomly downloaded from the internet, or purchased from an commercial entity. For pre-trained neural networks using proprietary training datasets with commercial secret or ethic/privacy concerns (such as commercial face datasets and CT scans from patients), the proposed method is still valid as long as there are are a few training samples for reference, or it is possible to request a few reference training samples.
>
> __Attack Detection for Federated Learning.__ In federated learning, raw training data (such as face images) is forbidden to be transmitted to the central server. And hence even the neural network trainer cannot access the full training dataset (will violate user privacy), and it is impossible to use any data-demanding methods to detect attack against a trained model (e.g., face recognition model). In contrast, the proposed method is still valid in this scenario as long as a few training samples can be collected from several volunteers for reference.
>
> We added these in the Appendix of the revised manuscript.
>
> **3. Performance when there is sufficient training data.**
>
> Please refer Section 5 “Training Set Size.” Our proposed method relies more on the stable pattern about model gradient consistency instead of the amount of data. It does not benefit from a large amount of data. According to our empirical observations, the performance gain from number of training data will become marginal starting from roughly 200 samples (L297-301), because the distribution of ARCv feature is already well-represented by the small batch of data. Meanwhile, there is no extra representational capacity of the easy-to-interpret 2-dimensional ARCv feature. Thus, our proposed method is more suitable for scenarios with extremely limited data.
>
> **4. Why use T=48 in L100?**
>
> In our experiments, we use T=6 (L181-184), and the corresponding features are visualized in Figure 2, Figure 3, and Figure 4. We speculate some readers might want to know what the feature wille be like with a larger step size. Thus, we show this to through T=48 examples in Figure 1, which meanwhile demonstrates that the network behaves more and more linear from step to step. With an empirically chosen T=48, the trend of the matrix is clear, and each cell in the matrix will not be too small to visualize. With a larger step size like T=100 or even larger, the matrix will show the same pattern, but the cells will be too small to visualize.

---

> > ### Comment · Reviewer_nyro · 2022-08-07
> > **Thanks for the response. I will keep my original score.**
> >
> > Thank the authors for the detailed response and comprehensive discussions. The proposed method is relatively new and interesting, which could potentially arouse future research along this line. Meanwhile, the application of the current method is somewhat limited as pointed out by the other reviewers. Therefore, I decided to keep my original score.

---

### Official Review · Reviewer_XmZm · 2022-07-12

**Rating:** 5
**Confidence:** 2
**Soundness:** 2 fair
**Presentation:** 3 good
**Contribution:** 3 good

**Summary:**

This paper proposed a method to trace the strong adversarial attacks (specifically PGD). The insight is PGD attack is likely to leave a trace and trigger the local linearity of the network. They then introduced the ARC to capture the SAE, resulting in a detector that can identify the informed attacks and uninformed attacks. Experiments including ImageNet showed the effectiveness.

**Questions:**

NA

**Strengths And Weaknesses:**


Strength:
1. It studies very extremely limited settings for the defender, which is novel and interesting.
2. The paper presents a host of visualization to support the intuitions and conclusions.

Comment:
1. My major concern is that ARC is not effective under a lot of scenarios (including FGSM, small $\epsilon$, and transfer-based attacks ). It is also possible for PGD attacks to bypass the detector by reducing the number of steps. It cannot work with a robust model (adv training). I am not sure if ARC is a valid trace method.

After rebuttal, I am willing to improve the score to 5, but my concern about the method still exists.

---

> ### Author Response · Authors · 2022-07-30
> **We explicitly justify limitations inherited from our assumptions. The proposed method is effective under clarified conditions.**
>
> **1. Is ARC a valid method?**
>
> ARC is effective and valid given the five assumptions are satisfied (See “Uniqueness of SAE to PGD-like Attack.” in Section 2). These assumptions collectively make the proposed method specific to PGD-like attacks, and hence is expectedly ineffective against non-PGD-like attacks. These assumptions are justified in Section 4.2 through both qualitative and quantitative demonstrations. Conclusively, SAE is a unique trace of PGD-like attacks (L269-L271), and is effective against PGD-like attacks (Section 4).
>
> We summarized a list of pros and cons of the proposed method in the Appendix of the revised manuscript.
>
> **2. Reducing number of steps for PGD-like attacks.**
>
> We agree with this. It is known that the number of iterations (fixed at 100 in our experiments) also impacts the attack strength besides perturbation magnitude ε. As increasing number of iterations will also lead to a more linear response from the model given an fixed and appropriate ε and achieve SAE similarly, we stick to one controlled variable ε for simplicity.
>
> On the contrary, reducing the number of iterations of a PGD-like attack will also lead to small perturbations that are hard to detect (as demonstrated in Section 4), and hence increase the possibility that the attack will not trigger clear SAE and hence bypass the proposed detection method. As an extreme case, FGSM, namely the single-step version of PGD does not effectively trigger SAE (as discussed in Section 4.2).
>
> The related works usually fix at a single set of attack parameters, and hence miss the observation that smaller perturbations are harder to detect.
>
> This is added the corresponding discussion in the Appendix (due to page limit) of the revised manuscript.
>
> **3. It cannot work with adversarial training.**
>
> The proposed method is incompatible with Adversarial Training. But meanwhile it provides a new perspective to understand why adversarial training works. See "Combination with Adversarial Training" in Section 5).

---

> > ### Comment · Reviewer_XmZm · 2022-08-05
> > **More Comments**
> >
> > I appreciate the author's response and am eager to discuss the paper.
> >
> > **ARC is effective and valid given the five assumptions are satisfied.**
> >
> > I agree that defending against or detecting adversarial attacks is exceptionally challenging, and some assumptions of the attackers are reasonable. However, I think the assumptions in this paper are too strong. The pipeline of the attack includes 1) attackers getting and using the victim's model parameters, 2) attackers choosing the exact same $L_p$ (not small $\epsilon$) bound with the defender,  3) attackers choosing the $L_{CE}$, 4) attackers using enough attacking iterations with PGD. I  think it is unlikely that the attackers can strictly follow them when constructing the adversarial examples.
> >
> > **Reducing the number of steps for PGD-like attacks.**
> >
> > Thank you for clarifying it. If SAE is not effective when facing weak adversarial examples (small $\epsilon$ and fewer attacking steps), then I think it is necessary to modify the first condition in Section 4.2 and discuss it.
> >
> > **Adaptive Attacks**
> >
> > Given those attacking assumptions, I think it might be unnecessary to even talk about adaptive attacks. Common adaptive attacks include:
> >
> > * Changing the loss function to attack the detector.
> > * Using BPDA to approximate the gradient.
> > * Evaluating with transfer-based attacks.
> > * Generating random noise with a large number of times.
> >
> > All of them are excluded from attacking assumptions. I am happy to talk more with the authors.

---

> > > ### Author Response · Authors · 2022-08-05
> > > **Further discussion**
> > >
> > > We appreciate the reviewer's additional comments. Here are our further discussions:
> > >
> > > **1. ARC is effective and valid given the five assumptions are satisfied.**
> > >
> > > **(1)** We acknowledge the mentioned strong assumptions imposed by our model. However, the assumptions that are uniquely specific to PGD-like attacks is meanwhile the reason why we can infer more information about the attack upon detection, e.g., Lp bound, loss function, and the label.
> > >
> > > **(2)** The related works focus on a wide range of attacks but can only infer whether the example is adversarial or not, while our method is specifically focusing on PGD-like attacks (namely the five assumptions), and can infer further information apart from whether the example is adversarial or not. As reflected by our title, we aim to reveal the unique trace (or “signature”) of the PGD-like attacks that are very popular among the attack/defense literature. We think identifying the unique trace of PGD-like also helps us to better understand the characteristics of PGD-like attacks as well as its difference from other attacks.
> > >
> > > **(3)** Our method is built upon a problem setting that requires less information from the potential users, which makes it usable in a wider range of potential scenarios. The methods that requires a large amount of training data will become infeasible when accessing a large amount of raw data is infeasible (e.g. Federated Learning as mentioned in the response to other reviewer comments. Accessing raw images from participating devices will violate user privacy).
> > >
> > > **2. Reducing the number of steps for PGD-like attacks.**
> > >
> > > We agree that clarifying the issue of number of steps makes the first condition more accurate. In the first revision of the manuscript we added Appendix A.2 to discuss the effect on number of steps and the expected results. We will conduct experiments with different numbers of steps on CIFAR-10 and add the experimental results later. We will eventually also add this discussion in Section 4.2 to make the condition less confusing and better justified.
> > >
> > > The preliminary revised condition (I) is as follows: “whether the input is adversarially perturbed by an iterative projected gradient update method with a not small number of steps.” And we will clarify in Section 4.2 that “the larger the number of steps is, the stronger the unique trace will be.”
> > >
> > > **3. Adaptive Attacks.**
> > >
> > > Apart from the adaptive attack that aims to make ARC ineffective (further discussed in Appendix A.4), other forms of adaptive attacks are also possible (further discussed in Appendix A.4 as well as the reply to reviewer vEc7 Based on “On Adaptive Attacks to Adversarial Example Defenses. Tramèr et al. NeurIPS 2020”).
> > >
> > > It is pointed out by reviewer vEc7 that our proposed method is spiritually similar to “The Odds are Odd” and “Turning a Weakness into a Strength” – both of them are statistical tests. Their corresponding adaptive attacks are “Logit Matching” and “Interpolation”, which is discussed in the response to reviewer vEc7 as well as Appendix A.4.
> > >
> > > As for the four kinds of adaptive attacks mentioned by the reviewer:
> > >
> > > **(1)** Changing the loss function to attack the detector. This is exactly breaking our condition (IV). With a different loss function, the SAE reflected by ARC feature will be weaker (discussed in Section 4.2).
> > >
> > > **(2)** Using BPDA to approximate the gradient. Our proposed method does not change the weights or architecture of a pre-trained neural network, and we can directly compute the gradient without gradient-masking effect. Hence it is unnecessary to approximate gradient and lower the success rate of attack.
> > >
> > > **(3)** We experiment with transfer-based attacks in Table 2 (t8: DI-FGSM and t9: TI-FGSM). Meanwhile, they break the condition (II) – if our proposed method does not raise alert, we will also know that the pre-trained model weights are probably not yet stolen by the attacker.
> > >
> > > **(4)** Generating random noise untill a successful attack is also a valid adaptive attack.
> > >
> > > In practice, breaking one of five conditions is simpler than directly attacking ARC. But we speculate that not discussing adaptive attack at all in the original submission will lead to a struggling rebuttal process.

---

> > > > ### Comment · Reviewer_XmZm · 2022-08-06
> > > > **Response to Authors**
> > > >
> > > > Thanks for answering my questions.
> > > >
> > > > I think the paper is very interesting and makes good contributions in general, especially for analyzing the PGD-like attacks.
> > > >
> > > > I agree that the method can infer more information and are data undemanding. However, failing to detect FGSM (the simplest white-box adversarial attack) and black-box attacks (the realistic setting for attackers) prevents it to deploy as a valid defense.
> > > >
> > > > In summary, I am willing to improve the score to borderline accept, since the authors clearly addressed all limitations in the paper. In addition, tracing the adversarial attacks is a promising research direction, and this paper can inspire more follow-up works.

---

> > > > > ### Author Response · Authors · 2022-08-06
> > > > > **Response to Reviewers**
> > > > >
> > > > > Thanks for the additional comments.
> > > > >
> > > > > As discussed with reviewer vEc7, the core part of the manuscript, as reflected by the title, is Section 2 which characterizes PGD-like attacks in non-adversarial setting. To quantitatively evaluate such characterization method, we use its direct application (attack detection) and presented Section 3.
> > > > >
> > > > > We will clearly justify the limitations of using such characterization for real detection as discussed with reviewers. Meanwhile we will make minor edits to make sure the emphasis put on attack detection does not outweigh the emphasis put on the non-adversarial characterization method itself.

---

> > > ### Author Response · Authors · 2022-08-06
> > > **Experiment with different number of steps**
> > >
> > > As discussed in the previous post ("2. Reducing the number of steps for PGD-like attacks."), we conduct experiments with different numbers of steps of BIM attack on CIFAR-10/ResNet-18, and report the corresponding results as follows:
> > >
> > > | Steps | DR   | FPR | Acc | Acc\* |
> > > |-------|------|-----|-----|-------|
> > > | 100   | 79.2 | 1.1 | 0.0 | 62.4  |
> > > | 50    | 75.0 | 1.1 | 0.0 | 58.1  |
> > > | 25    | 64.1 | 1.1 | 0.0 | 47.3  |
> > > | 15    | 49.3 | 1.1 | 0.0 | 33.5  |
> > > | 10    | 33.1 | 1.1 | 0.2 | 20.1  |
> > > | 08    | 22.4 | 1.1 | 0.7 | 12.2  |
> > > | 05    |  7.1 | 1.1 | 3.7 |  5.5  |
> > >
> > > Which is consistent with our expectation. We will add this to the appendix and justify its effect in Section 4.2.

---

### Meta-Review · Area_Chair_P7Pf · 2022-08-24

**Recommendation:** Reject
**Confidence:** Certain

**Metareview:**

This paper observes that "PGD-like" attack algorithms have characteristics that
allow one to detect an input has been attacked. While I agree that it is
interesting PGD-like attacks have detectable properties, I agree with the
reviewers that designing attck-specific defenses has limited utility. The
authors already show that FGSM and similar attacks are not detected with this
approach, and this makes me worry that adaptive attacks will also not be
easy to detect. And so while making an observation about how PGD works is
interesting, it is not yet sufficient and will likely not form the basis of
a strong defense.


**Award:**

No

---

### Decision · Program_Chairs · 2022-09-14

Reject